



# Load Reduction for Wind Turbines: an Output Constrained, Subspace Predictive Repetitive Control Approach

Yichao Liu[1], Riccardo Ferrari[1], and Jan-Willem van Wingerden[1]

[1]Delft Center for Systems and Control, Delft University of Technology, Delft, The Netherlands

**Correspondence:** Yichao Liu (yichaoliu629@gmail.com)

**Abstract.** Individual Pitch Control (IPC) is a well-known approach to reduce blade loads on wind turbines. Although very effective, IPC usually requires high levels of actuator activities, which significantly increases the pitch Actuator Duty Cycle (ADC). This will subsequently result in an increase of the wear on the bearings of the blades and make the current IPC design not economical viable. An alternative approach to this issue is to reduce the actuator activities by incorporating the output constraints in IPC. In this paper, a fully data driven IPC approach, which is called constrained Subspace Predictive Repetitive Control (cSPRC) is introduced. The output constraints can be explicitly considered in the control problem formulation via a Model Predictive Control (MPC) approach. The cSPRC approach will actively produce the IPC action for the necessary load reduction when the blade loads violate the output constraints. In this way, actuator activities can be significantly reduced. Two kinds of scenarios are simulated to illustrate the unique applications of the proposed method: wake-rotor overlap and turbulent wind conditions. Simulation results show that the developed cSPRC is able to account for the output constraints into the control problem formulation. Since the IPC action from cSPRC is only triggered to prevent violating the output constraints, the actuator activities are significantly reduced. This will help to reduce the pitch ADC, thus leading to an economical viable load control strategy. In addition, this approach allows the wind farm operator to design conservative bounds to guarantee the safety of the wind turbine control system.

## 1 Introduction

Over the past decades, wind energy has expanded by leaps and bounds in the international energy mix (Watson et al., 2019). In total, 60.4GW global wind energy capacity was installed in 2019, which shows a rapid growth of 19% compared to 2018 (Lee and Zhao, 2020).

However, one of the main challenges in the development of wind farms is the high Operation & Maintenance (O&M) cost (Willis et al., 2018). This is usually related to the design of the wind turbine which is subjected to severe dynamic loading. In particular, in a wind farm downstream turbines will be affected by the wake flow of the neighboring upstream turbines. The interaction between wake and turbines will lead to increased blade loads and loss of power (Frederik et al., 2020). In addition, atmospheric turbulence will have a negative impact on the wind turbine performance (Barthelmie et al., 2007). Since wind turbines tend to have larger rotor diameters and a more slender tower than land-based counterpart, the effects of these dynamic loading would be more significant. Therefore, load mitigation concerning the wind turbines erected in a wind farm becomes of





vital importance to guarantee the reliability of the turbine system and to reduce the O&M costs (Njiri and Söffker, 2016). In general, the majority of the loads on wind turbine rotors shows a periodic nature (Yuan et al., 2020). Individual Pitch Control (IPC) has demonstrated its effectiveness in reduction of these periodic loads (Bossanyi, 2003). In IPC, the pitch angle of each blade is regulated independently with the aid of individual pitch actuators and measurements of the bending moments. By
superimposing the periodic pitch angles to each blade on top of the collective pitch, the blade loads can be alleviated.

Bossanyi (Bossanyi, 2003) initially demonstrated the possibility of reducing the blade loads occurring at an angular frequency of Once Per (1P) rotation, by using an IPC based on a Linear Quadratic Gaussian (LQG) approach. However, the 1P loads are symmetric and thus are not the dominant loads on the non-rotating components of the wind turbine. These components suffer from the largest loads at the blade passing frequency $N$P (3P for a three-bladed wind turbine). Therefore, the
Coleman transformation (Bir, 2008), which converts the loads from the rotating frame of reference into the static frame, was suggested. This makes it possible to use simple linear Single-Input Single-Output (SISO) control approaches for IPC, such as Proportional–Integral (PI) controllers (Bossanyi, 2005; van Solingen et al., 2014). More recently, other advanced IPC approaches, such as fixed-structure $H_\infty$ feedback-feedforward IPC (Ungurán et al., 2019) and multivariable robust IPC (Yuan et al., 2020), were developed to mitigate the blade loads on the wind turbine. On the other hand, the application of IPC to wake
load control is receiving increasing attention. Yang *et al.* (Yang et al., 2011) developed a periodic IPC to deal with the blade loads of the wind turbine in a wind farm where the wake interactions are considered using Jensen and Larsen wake models. However, the wind farm wake shows a more challenging characteristic, namely the wake meandering (Larsen et al., 2008), which was not considered in the developed approach. In order to address the challenge of such a complex wake meandering phenomenon, a new IPC, which is based on a multiple Model Predictive Control (MPC) approach, was proposed by Yang *et*
*al.* (Yang et al., 2015).

The drawback of these approaches is that the pitch Actuator Duty Cycle (ADC) is dramatically increased due to the cyclic fatigue loads on the pitch actuators. Such an effect is worsened when these approaches attempt to control the non-deterministic wind loads at high wind turbulence intensities, and the dynamic loading caused by the wake. This will result in an increase of the wear on the bearings of the blades and eventually a shortening of the lifespan of the pitch control system. Moreover, the pitch
control system is usually subjected to various constraints due to the physical restrictions of the pitch actuator, safety limitations, environmental regulations and wind farm manufacturer specifications (Vali et al., 2016). Exceeding these constraints may result in damage to the pitch control system and ultimately in the failure of the entire wind turbine.

In order to address this challenge, a constrained IPC was recently developed by Petrović *et al.* (Petrović et al., 2020). In their work, the input constraints of the pitch actuators are explicitly taken into account by using an MPC framework,
which makes it possible to reduce the actuator activities. However, output constraints, to the best of author's knowledge, have not been investigated yet for the case of IPC. As a control system with output constraints is capable of limiting the loads within certain safety bounds, it would be an ideal way to reduce the actuator activities. Compared to the widely-used input constraints, the inclusion of output constraints is more challenging. It may induce an unstable, closed-loop system even though the corresponding unconstrained algorithm is stable (Wang, 2009).



In order to approach the goal of introducing output constraints in wind turbine control, a novel IPC approach is presented in this paper. It is based on a constrained Subspace Predictive Repetitive Control (cSPRC). The basic concept of SPRC was initially proposed by van Wingerden *et al.* (van Wingerden et al., 2011). It is essentially a fully data-driven approach comprised of subspace identification and repetitive control. The subspace identification step, based on an online solution, is used to recursively derive a linear approximation of the wind turbine dynamics (van der Veen et al., 2013). Based on it, a predictive repetitive control law is then formulated to reduce only specific deterministic loads, such as 1P loads, under varying operating conditions. The SPRC approach has shown promising results in numerical simulations (Navalkar et al., 2014; Liu et al., 2020) and in wind tunnel experiments (Navalkar et al., 2015; Frederik et al., 2018).

The main contributions of this paper lie in the following two aspects. The first contribution is the data-driven framework. For the first time, the constraints of the control problem, especially the output constraints of the blade loads, are explicitly considered in the repetitive control formulation. This is achieved by integrating an MPC approach (Qin and Badgwell, 2003) into SPRC, so that the repetitive control law subjected to specified output constraints can be formulated. Since the accuracy of the control output prediction is affected by the model uncertainty of the identified model, the output constraints may cause instability of the closed-loop system and severe deterioration of the control performance (Wang, 2009). Therefore, the output constraints are implemented as soft constraints by introducing so-called slack variables in the control problem formulation. The output constraints will be relaxed if the slack variables become large enough. In case there are no constraint violations, only the widely-used baseline pitch controller is active to maintain basic wind turbine performance. Once the blade loads induced by the wind turbulence and wind farm wake increase and violate the output constraints, cSPRC will actively produce the IPC action for the necessary load mitigation. This is achieved by penalising the control inputs only in the control objectives, which ensures that the controller will be inactive if the blade loads are lower than the safety bounds. Moreover, the safety bounds, corresponding to the values of the output constraints in cSPRC, can be designed according to the design regulations of wind farms, such as IEC 61400-1 (IEC, 2005). Since cSPRC is only enabled for necessary load reduction, the pitch activities would be significantly reduced while the safety of the wind turbine can be still guaranteed.

The second contribution is the unique application of the cSPRC approach to two independent scenarios: one where the wind turbine is impinged and overlapped by the wake shed from the upstream turbine and one where turbulent wind flow is present, respectively. In particular, in the wind farm wake scenario the wind turbine will experience partial and full wake overlap due to the wind direction change and the yaw misalignment of the upstream turbine (Fleming et al., 2015). The partial wake overlap, together with the velocity deficit within the wake, will induce asymmetric loading of the rotor plane of the wind turbine. This will vividly illustrate the capability of dealing with output constraints in the cSPRC approach.

The effectiveness of the cSPRC approach under these two typical scenarios will be demonstrated through high-fidelity simulations. For this, the FLOw Redirection and Induction in Steady-state (FLORIS) model (Gebraad et al., 2016), which is a parametric model for predicting a steady-state wake in a wind farm, is utilized to simulate the wind farm wake. It actually provides the wind speed input for the wind turbine simulations. Then, the wind turbine simulations are executed using the U.S. National Renewable Energy Laboratory (NREL)'s Fatigue, Aerodynamics, Structures, and Turbulence (FAST) tool (Jonkman and Buhl, 2005). In this respect, a 10MW wind turbine model is used, which is developed by the Technical University of Den-



mark (DTU) (Bak et al., 2013) and the Stuttgart Wind Energy (SWE) institute (Lemmer et al., 2016). A thorough comparison against baseline and conventional IPC approaches is made to evaluate the performance of the proposed cSPRC.

The remainder of this paper is organized as follows. Section 2 introduces the wind turbine model and the simulation environment. In Section 3, the methodology of the cSPRC with the inclusion of output constraints is elaborated. Then, the potential of cSPRC for load mitigation in wake-rotor overlap and turbulent wind scenarios is illustrated through high-fidelity simulations in Section 4. Conclusions are drawn and future work is discussed in Section 5.

## 2   Wind turbine model and simulation environment

In this section, the wind turbine model and its simulation environment are introduced. The wind turbine model is based on the DTU 10MW three-bladed variable speed reference wind turbine. Its specifications are presented in Table 1. More details can be found in the reports (Bak et al., 2013; Lemmer et al., 2016).

**Table 1.** Specification of the DTU 10MW reference wind turbine model.

| Parameter | Value |
| --- | --- |
| Rating | 10 MW |
| Rotor orientation, configuration | Upwind, 3 blades |
| Pitch control | Variable speed, baseline and IPC |
| Drivetrain | Medium speed, multiple stage gearbox |
| Rotor, hub diameter | 178.3 m, 5.6 m |
| Hub height | 119 m |
| Cut-in, rated, cut-out wind speed | 4 m/s, 11.4 m/s, 25 m/s |
| Cut-in, rated rotor speed | 6 rpm, 9.6 rpm |
| Rated tip speed | 90 m/s |

Based on the wind turbine model, the implementation of the case study is illustrated in Figure 1. The aero-structural dynamic part of the wind turbine is simulated in the FAST model (Jonkman and Buhl, 2005) while the turbine control part is implemented in *Simulink* (Mulders et al., 2019). The developed the cSPRC approach, which is encompassed by a light grey block, will be described in Section 3. Other two pitch control strategies are implemented for comparisons. They are: 1) Baseline control based on the Collective Pitch Control (CPC) approach (Jonkman and Buhl, 2005). In CPC, the classical gain-scheduled Proportional Integral (PI) control (Boukhezzar et al., 2007) is utilized to regulate the pitch angles of all blades synchronously. It is denoted by a white block in Figure 1. 2) Conventional IPC, which is based on the Multi-Blade Coordinate (MBC)-based IPC. In MBC-based IPC, the pitch angle of each blade is regulated independently with the aid of the so-called Coleman transformation (Bir, 2008). Note that the constraints are usually not considered in such an MBC-based IPC approach (Selvam et al., 2009). It is indicated by a dark grey block in Figure 1.





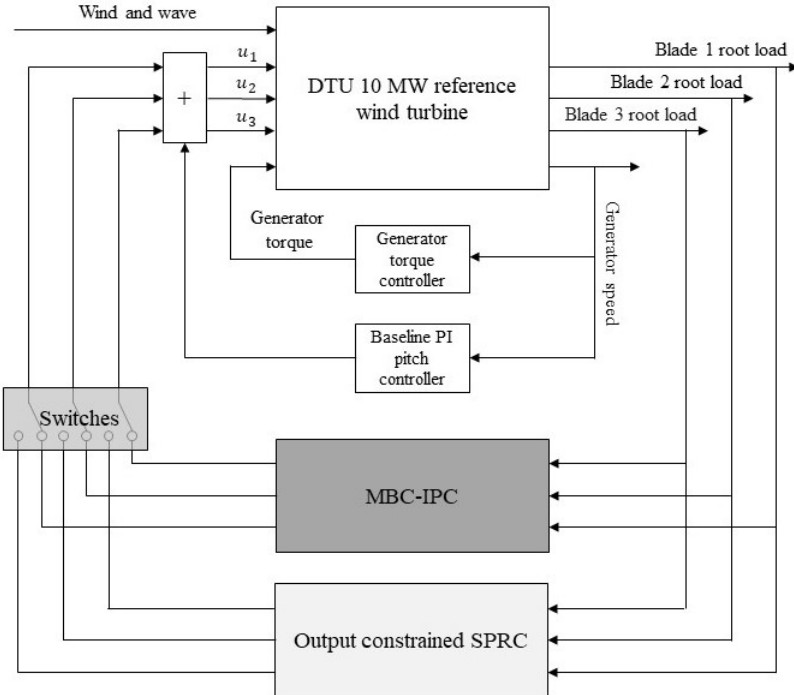

**Figure 1.** Block diagram of the wind turbine model and of the control loop. The aero-hydro-structural dynamic part is simulated in FAST code while the turbine control part including a baseline CPC (Jonkman and Buhl, 2005), an MBC-IPC (Mulders et al., 2019) and the proposed cSPRC, are implemented in Simulink. The two IPC controllers can be selectively enabled in order to compare their performances against each other and against the baseline controller alone. The baseline PI pitch controller and generator torque controller are always activated to guarantee the basic performance of the wind power generation.

The baseline control is based on a Linear Time Invariant (LTI) dynamical system (Bak et al., 2013). The MBC-based IPC is implemented following the work of Multders *et al.* (Mulders et al., 2019). On the other hand, the proposed cSPRC approach, will be introduced in Section 3 to show the capability of output constraints and its application to the load control of wake overlapping and turbulent wind flow scenarios.

## 3   Output constrained, subspace predictive repetitive control

This section outlines the theoretical framework of the cSPRC approach for wind turbine load control. First of all, a discrete-time LTI system along with an output predictor is established to approximate the wind turbine dynamics. All the parameters of the linear representation are then identified via an online recursive subspace identification. Based on this, the predictive repetitive control law subjected to the different kinds of constraints is then synthesized by solving an MPC optimal problem in receding horizon. Especially, the output constraints of the controller, because of the presence of uncertainty in the identified model, are implemented as soft constraints by introducing slack variables in the MPC. Furthermore, only the control inputs in





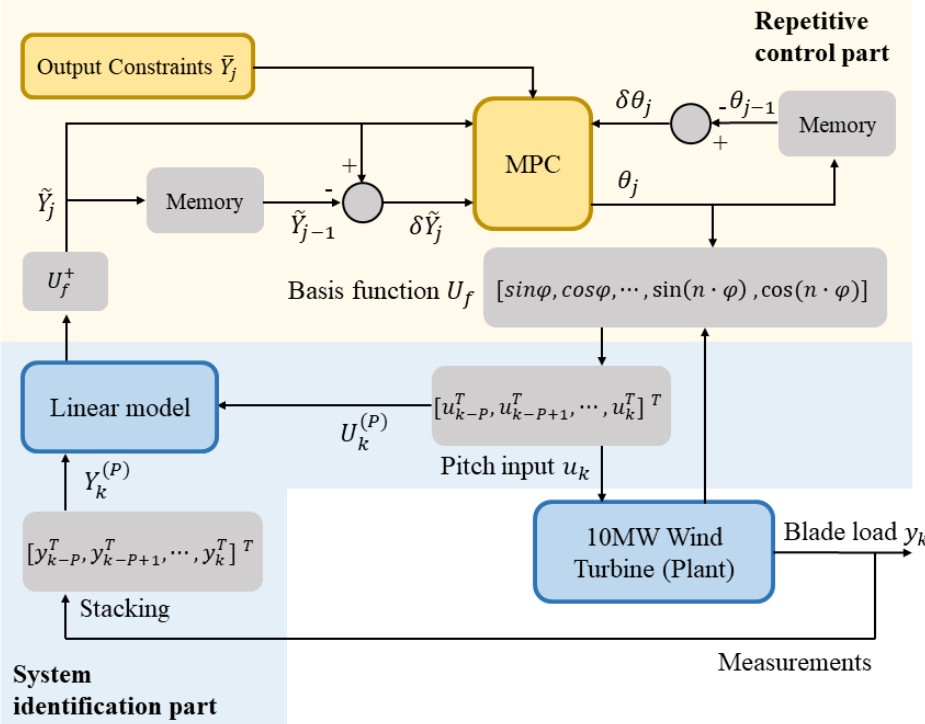

**Figure 2.** Implementation of cSPRC, which includes online system identification and repetitive control. MPC optimization is used to incorporate the output constraints in repetitive control formulation.

the MPC are penalised, which ensures that the controller will be only activated for load mitigation when the blade loads violate the output constraints. The overall structure of the cSPRC approach has been illustrated in Figure 2.

### 3.1 Online recursive subspace identification

In the cSPRC framework, the wind turbine dynamics are represented by an LTI system affected by unknown periodic distur-
bances (Houtzager et al., 2013) as

$$
\begin{cases}
x_{k+1} & = Ax_k + Bu_k + Ed_k + Le_k \\
y_k & = Cx_k + Fd_k + e_k
\end{cases},
\tag{1}
$$

where $x_k \in \mathbb{R}^n$, $u_k \in \mathbb{R}^r$ and $y_k \in \mathbb{R}^l$ denote the state, control input and output vectors. In the wind turbine model, $r = l = 3$. $u_k$ and $y_k$ represent the blades pitch angles and the blade loads, i.e. Out-of-Plane bending moment (MOoP), respectively at discrete time index $k$. Moreover, $d_k \in \mathbb{R}^m$ is the periodic disturbance component of the loads at the blade root, while $e_k \in \mathbb{R}^l$ is
the zero-mean white innovation process or the aperiodic component of the blade loads. Other matrices $A \in \mathbb{R}^{n \times n}$, $B \in \mathbb{R}^{n \times r}$,



$C \in \mathbb{R}^{l \times n}$, $L \in \mathbb{R}^{n \times l}$, $E \in \mathbb{R}^{n \times m}$ and $F \in \mathbb{R}^{l \times m}$ denote the state transition, input, output, observer, periodic noise input and periodic noise direct feed-through matrices, respectively.

The following equations can be derived by rewriting (1) in predictor form

$$
\begin{cases}
x_{k+1} & = \tilde{A}x_k + Bu_k + \tilde{E}d_k + Ly_k, \\
y_k & = Cx_k + Fd_k + e_k
\end{cases}
\tag{2}
$$

in which $\tilde{A} \triangleq A - LC$ and $\tilde{E} \triangleq E - LF$. Let us define a periodic difference operator $\delta x_k \triangleq x_k - x_{k-P}$, where $P$ is the period of the disturbance, equalling to the blade rotation period. Then the effect of the periodic blade loads $d_k$ on the input-output system can be eliminated as it holds

$$
\delta d_k = d_k - d_{k-P} = 0.
$$

Similarly, $\delta u$, $\delta u$ and $\delta e$ can be defined as well. Applying the $\delta$-notation to (2), this equation can be rewritten as follows,
where the periodic blade load term disappears.

$$
\begin{cases}
\delta x_{k+1} & = \tilde{A}\delta x_k + B\delta u_k + L\delta y_k \\
\delta y_k & = C\delta x_k + \delta e_k.
\end{cases}
\tag{3}
$$

Then, a stacked vector $\delta U_k^{(p)}$ for a past time window with the length of $p$ is defined as

$$
\delta U_k^{(p)} = \begin{bmatrix} u_k - u_{k-P} \\ u_{k+1} - u_{k-P+1} \\ \vdots \\ u_{k+p-1} - u_{k+p-P-1} \end{bmatrix}.
\tag{4}
$$

Similarly, the vector $\delta Y_k^{(p)}$ is defined. Next, the future state vector $\delta x_{k+p}$ is introduced based on $\delta U_k^{(p)}$ and $\delta Y_k^{(p)}$ as

$$
\delta x_{k+p} = \tilde{A}^p \delta x_k + \begin{bmatrix} K_u^{(p)} & K_y^{(p)} \end{bmatrix} \begin{bmatrix} \delta U_k^{(p)} \\ \delta Y_k^{(p)} \end{bmatrix},
\tag{5}
$$

in which $K_u^{(p)}$ and $K_y^{(p)}$ are:

$$
K_u^{(p)} = \begin{bmatrix} \tilde{A}^{p-1}B & \tilde{A}^{p-2}B & \cdots & B \end{bmatrix},
$$
$$
K_y^{(p)} = \begin{bmatrix} \tilde{A}^{p-1}L & \tilde{A}^{p-2}L & \cdots & L \end{bmatrix}.
$$

It is worth noting that $p$ need to be selected large enough, which guarantees $\tilde{A}^j \approx 0 \ \forall j \geq p$ (Chiuso, 2007). With this in
mind, $\delta x_{k+p}$ can be simplified as

$$
\delta x_{k+p} \approx \begin{bmatrix} K_u^{(p)} & K_y^{(p)} \end{bmatrix} \begin{bmatrix} \delta U_k^{(p)} \\ \delta Y_k^{(p)} \end{bmatrix}.
\tag{6}
$$



By substituting this equation into (3), the approximation of $\delta y_{k+p}$ is derived as

$$\delta y_{k+p} \approx \underbrace{\left[ \begin{array}{cc} CK_u^{(p)} & CK_y^{(p)} \end{array} \right]}_{\Xi} \left[ \begin{array}{c} \delta U_k^{(p)} \\ \delta Y_k^{(p)} \end{array} \right] + \delta e_{k+p}. \tag{7}$$

From (7), it can be seen that the matrix of coefficients $\left[ \begin{array}{cc} CK_u^{(p)} & CK_y^{(p)} \end{array} \right]$, which is the so-called Markov matrix $\Xi$, includes

all the necessary information on the behaviour of the wind turbine system. It is determined by the input vector $u^{(r)}$ and output vector $y^{(l)}$. In essence, the subspace identification aims to find an online solution of the following Recursive Least-Squares (RLS) optimization problem (van der Veen et al., 2013)

$$\hat{\Xi}_k = \arg\min_{\hat{\Xi}_k} \sum_{i=-\infty}^{k} \left\| \delta y_i - \lambda \hat{\Xi}_k \left[ \begin{array}{c} \delta U_{i-p}^{(p)} \\ \delta Y_{i-p}^{(p)} \end{array} \right] \right\|_2^2. \tag{8}$$

In (8), $\lambda$ is a forgetting factor ($0 \ll \lambda \le 1$) to alleviate the effect of past data, and adapt to the updated system dynamics online.

In this paper, a value close to 1, i.e. $\lambda = 0.9999$, is chosen to guarantee the robustness of the optimization process. According to the definition of $\Xi$ in (7), $\hat{\Xi}_k$ at time index $k$ includes estimates of the following matrices,

$$\hat{\Xi}_k = \left[ \begin{array}{ccccccccc} \widehat{C\tilde{A}^{p-1}B} & \widehat{C\tilde{A}^{p-2}B} & \cdots & \widehat{CB} & \widehat{C\tilde{A}^{p-1}K} & \widehat{C\tilde{A}^{p-2}K} & \cdots & \widehat{CK} \end{array} \right]. \tag{9}$$

To obtain a unique solution to this RLS optimization problem, a persistent exciting signals is superimposed on the top of the control input. Subsequently, this RLS optimization (8) is implemented with a QR algorithm (Sayed and Kailath, 1998) in an

online recursive manner to obtain $\hat{\Xi}_k$. The estimates of $\hat{\Xi}_k$ are then used in an MPC framework to formulate a repetitive control law subjected to the output constraints. The implementation of the repetitive control problem formulation in MPC, namely the receding horizon repetitive control, will be elaborated in the next subsection.

### 3.2 Output constrained repetitive control

Based on the LTI system obtained in subspace identification step, the constrained repetitive control law is formulated over a

period $P$. Considering that $P \ge p$ and usually $P$ is much larger than $p$, the output equation can be lifted over the period $P$ as

$$\delta Y_{k+p}^{(P)} = \tilde{\Gamma}^{(P)} \delta x_{k+p} + \left[ \begin{array}{cc} \tilde{H}^{(P)} & \tilde{G}^{(P)} \end{array} \right] \left[ \begin{array}{c} \delta U_{k+p}^{(P)} \\ \delta Y_{k+p}^{(P)} \end{array} \right]. \tag{10}$$



$\tilde{H}^{(P)}$ is the Toeplitz matrix, which is defined as

$$
\tilde{H}^{(P)} =
\begin{bmatrix}
0 & 0 & 0 & \cdots \\
CB & 0 & 0 & \cdots \\
C\tilde{A}B & CB & 0 & \cdots \\
\vdots & \vdots & \ddots & \vdots \\
C\tilde{A}^{p-1}B & C\tilde{A}^{p-2}B & C\tilde{A}^{p-3}B & \cdots \\
0 & C\tilde{A}^{p-1}B & C\tilde{A}^{p-2}B & \cdots \\
0 & 0 & C\tilde{A}^{p-1}B & \cdots \\
\vdots & \vdots & \ddots & \ddots
\end{bmatrix}.
\tag{11}
$$

By replacing $B$ with $L$, $\tilde{G}^{(P)}$ can be derived as well. In addition, the extended observability matrix $\tilde{\Gamma}^{(P)}$ is defined as

180

$$
\tilde{\Gamma}^{(P)} =
\begin{bmatrix}
C \\
C\tilde{A} \\
C\tilde{A}^2 \\
\vdots \\
C\tilde{A}^p \\
0 \\
\vdots \\
0
\end{bmatrix}.
\tag{12}
$$

Substituting (10) with (6), it yields,

$$
\delta Y_{k+P}^{(P)} = \tilde{\Gamma}^{(P)} \begin{bmatrix} K_u^{(P)} & K_y^{(P)} \end{bmatrix} \begin{bmatrix} \delta U_k^{(P)} \\ \delta Y_k^{(P)} \end{bmatrix} + \begin{bmatrix} \tilde{H}^{(P)} & \tilde{G}^{(P)} \end{bmatrix} \begin{bmatrix} \delta U_{k+P}^{(P)} \\ \delta Y_{k+P}^{(P)} \end{bmatrix}.
\tag{13}
$$

In (13), it is worth noting that the first $(P-p) \cdot r$ columns of $K_u^{(P)}$ and $K_y^{(P)}$ are 0. Moreover, the future output $\delta Y_{k+P}^{(P)}$ is actually predicted by previous $\delta Y_k^{(P)}$ and $\delta U_k^{(P)}$ and future input $\delta U_{k+P}^{(P)}$. It can be rewritten as


$$
\delta Y_{k+P}^{(P)} = \begin{bmatrix} \widehat{\Gamma^{(P)} K_u^{(P)}} & \widehat{\Gamma^{(P)} K_y^{(P)}} \end{bmatrix} \begin{bmatrix} \delta U_k^{(P)} \\ \delta Y_k^{(P)} \end{bmatrix} + \hat{H}^{(P)} \delta U_{k+P}^{(P)},
\tag{14}
$$

with the definitions of $\Gamma^{(P)}$ and $\hat{H}^{(P)}$ as follows,

$$
(I - \tilde{G}^{(P)})^{-1} \tilde{\Gamma}^{(P)} = \Gamma^{(P)}
$$

$$
(I - \tilde{G}^{(P)})^{-1} \tilde{H}^{(P)} = \hat{H}^{(P)}.
$$





In order to take into account output $Y_k^{(P)}$ in the optimization problem, (14) can be expanded as

$$
\quad Y_{k+P}^{(P)} = \left[\begin{array}{ccc} I_{l \cdot P} & \widehat{\Gamma^{(P)} K_u^{(P)}} & \widehat{\Gamma^{(P)} K_y^{(P)}} \end{array}\right] \left[\begin{array}{c} Y_k^{(P)} \\ \delta U_k^{(P)} \\ \delta Y_k^{(P)} \end{array}\right] + \hat{H}^{(P)} \delta U_{k+P}^{(P)} .
\tag{15}
$$

In order to incorporate the output constraints into the control problem formulation, an MPC controller is subsequently implemented to synthesize the repetitive control law. With this in mind, (15) is reformulated into a state space representation where the synthesized final input can be penalised as well, which is

$$
\underbrace{\left[\begin{array}{c} Y_{k+P}^{(P)} \\ \delta U_{k+P}^{(P)} \\ \delta Y_{k+P}^{(P)} \\ U_{k+P}^{(P)} \end{array}\right]}_{\hat{K}_{k+P}} = \underbrace{\left[\begin{array}{cccc} I_{l \cdot P} & \widehat{\Gamma^{(P)} K_u^{(P)}} & \widehat{\Gamma^{(P)} K_y^{(P)}} & -\hat{H}_k^{(P)} \\ 0_{(r \cdot P) \times (l \cdot P)} & 0_{r \cdot P} & 0_{(r \cdot P) \times (l \cdot P)} & -I_{r \cdot P} \\ 0_{l \cdot P} & \widehat{\Gamma^{(P)} K_u^{(P)}} & \widehat{\Gamma^{(P)} K_y^{(P)}} & -\hat{H}_k^{(P)} \\ 0_{(r \cdot P) \times (l \cdot P)} & 0_{r \cdot P} & 0_{(r \cdot P) \times (l \cdot P)} & 0_{r \cdot P} \end{array}\right]}_{\hat{A}_k} \underbrace{\left[\begin{array}{c} Y_k^{(P)} \\ \delta U_k^{(P)} \\ \delta Y_k^{(P)} \\ U_k^{(P)} \end{array}\right]}_{\hat{K}_k} + \underbrace{\left[\begin{array}{c} \hat{H}^{(P)} \\ I_{r \cdot P} \\ \hat{H}^{(P)} \\ I_{r \cdot P} \end{array}\right]}_{\hat{B}_k} U_{k+P}^{(P)} .
\tag{16}
$$

The state transition and input matrices are updated at each discrete time instance $k$. Next, a basis function projection (van Wingerden et al., 2011) is employed to limit the spectral content of the pitch control input within the frequency range of interest. More importantly, it will reduce the dimension of (16) that must be solved in the MPC framework, thus leading to a reduced computational cost. In this paper, the 1P frequency load on the rotor blades, which are mainly induced by the wind shear, wind turbulence, changes in the inflow wind speed and tower shadow, are taken into account. The transformation matrix
of the basis function projection can thus be defined as

$$
\phi = \underbrace{\left[\begin{array}{cc} \sin(2\pi/P) & \cos(2\pi/P) \\ \sin(4\pi/P) & \cos(4\pi/P) \\ \vdots & \vdots \\ \sin(2\pi) & \cos(2\pi) \end{array}\right]}_{U_f} \otimes I_r ,
\tag{17}
$$

where $U_f$ is the so-called basis function. The mathematical symbol $\otimes$ represents the Kronecker product. Note that the rotor azimuth $\psi$, equal to $2\pi k/P$ at each time step $k$, is used in the implementation, instead of the fixed $U_f$. This makes it possible to take into consideration the rotor speed variations caused by wind turbulence, or by changes of the inflow wind speed. Based on
the basis function, the control inputs at specific frequencies can be synthesized by taking a linear combination of the sinusoids of the transformation matrix as

$$
U_k^{(P)} = \phi \cdot \theta_j ,
\tag{18}
$$

where $j = 0, 1, 2, \cdots$ represents the rotation count, while $\theta \in \mathbb{R}^{4r}$, which determines the amplitudes and phase of the sinusoids, is computed based on (16) at each $P$. Similarly, the output can be transformed onto the subspace that defined by the basis





function, as

$$\tilde{Y}_j = \phi^+ Y_k^{(P)},\tag{19}$$

in which the symbol $+$ denotes the Moore-Penrose pseudo-inverse. Based on the basis function projection, (16) is reduced into a lower dimensional form as

$$
\underbrace{\begin{bmatrix} \tilde{Y}_{j+1} \\ \delta\theta_{j+1} \\ \delta\bar{Y}_{j+1} \\ \theta_{j+1} \end{bmatrix}}_{\bar{K}_{j+1}} = \underbrace{\begin{bmatrix} I_{l\cdot P} & \phi^+\widehat{\Gamma^{(P)}K_u^{(P)}}\phi & \phi^+\widehat{\Gamma^{(P)}K_y^{(P)}}\phi & -\phi^+\hat{H}^{(P)}\phi \\ 0_{r\cdot P\times l\cdot P} & 0_{r\cdot P} & 0_{r\cdot P\times l\cdot P} & -I_{r\cdot P} \\ 0_{l\cdot P} & \phi^+\widehat{\Gamma^{(P)}K_u^{(P)}}\phi & \phi^+\widehat{\Gamma^{(P)}K_y^{(P)}}\phi & -\phi^+\hat{H}^{(P)}\phi \\ 0_{r\cdot P\times l\cdot P} & 0_{r\cdot P} & 0_{r\cdot P\times l\cdot P} & 0_{r\cdot P} \end{bmatrix}}_{\bar{A}_j} \underbrace{\begin{bmatrix} \tilde{Y}_j \\ \delta\theta_j \\ \delta Y_j \\ \theta_j \end{bmatrix}}_{\bar{K}_j} + \underbrace{\begin{bmatrix} \phi^+\hat{H}^{(P)}\phi \\ I_{r\cdot P} \\ \phi^+\hat{H}^{(P)}\phi \\ I_{r\cdot P} \end{bmatrix}}_{\hat{B}_j} \theta_{j+1}.\tag{20}
$$

Compared to (16), the dimension of the projected matrix, i.e., $\bar{A} \in \mathbb{R}^{12l\times12l}$, is much lower than the original matrix $\hat{A} \in \mathbb{R}^{((2l+r)\cdot P)\times((2l+r)\cdot P)}$. Considering $P \gg 2r$, the order of the state-space representation as well as the following MPC optimization problem can be substantially reduced. On the other hand, such a basis function transformation guarantees that the input $U_k^{(P)}$ is a smooth signal at the specific frequencies. Then, the following output constraints, considering the transformation in (19), are imposed on (20) for all $j \geq 0$ as

$$Y_{k+P}^{(P)} \leq \bar{Y},\tag{21}$$

where $\bar{Y}$ is the constraints of the future output $Y_{k+P}^{(P)}$, which corresponds to the designed safety bounds of the blade loads. If the blade loads do not exceed the desired bounds, then the safety of the rotor is guaranteed. The value of the bounds can be determined by the wind farm operator or according to the safety factors of the loads in the design regulation such as IEC 61400-1 (IEC, 2005), or other safety limitations and environmental conditions. In this paper, $\bar{T}$ is calculated according to the IEC regulation (IEC, 2005). Since the future output $\tilde{Y}_{j+i|j}$ corresponds to the first element in (20), such output constraints can be converted into input constraints. Substituting (20) into (21), the output constraints are reformulated as

$$(\phi\cdot C\cdot \bar{B}_j)\cdot\theta_{j+1} \leq \bar{Y} - \phi\cdot C\cdot\bar{A}_j\bar{K}_j,\tag{22}$$

where $C = \text{diag}(I_{l\cdot P}, 0_{r\cdot P}, 0_{l\cdot P}, 0_{r\cdot P})$. Following the philosophy of the MPC algorithm, the control objectives are introduced in the following cost function as

$$J(\bar{K}, \mathbf{U}) = \sum_{i=0}^{N_p} (\bar{K}_{j+i|j})^T Q\bar{K}_{j+i|j} + \sum_{i=1}^{N_u} (\theta_{j+i|j})^T R\theta_{j+i|j},\tag{23}$$

with the MPC optimization problem as

$$V(\bar{K}_j) = \min_{\mathbf{U}} J(\bar{K}_j, \mathbf{U}),\tag{24}$$





subjected to

$$(\phi \cdot C \cdot \bar{B}_{j+i-1}) \cdot \theta_{j+i} \leq \bar{Y} - \phi \cdot C \cdot \bar{A}_{j+i-1} \bar{K}_{j+i-1}, \quad i = 1, \cdots, N_u, \tag{25}$$

where $Q$ and $R$ denote the positive-definite weighting matrices, while $N_p$ and $N_u$ are the prediction and control horizons, respectively. Since the controller is only active when the blade loads violate the output constraints, $Q$ is set to an all-zeros weight matrix. This will make $\theta_{j+i|j}$ the only penalization term in the cost function. $\mathbf{U} \triangleq [\theta_{j+1}^T, \cdots, \theta_{j+N_u}^T] \in \mathbb{R}^{4r \cdot N_u}$ is a sequence of a series of future control actions. They are computed by the MPC optimization over the prediction horizon at each rotation count $j$. This will optimize the future behavior of the wind turbine while respecting the output constraints in (21).

As usual in MPC implementations, only the first element $\theta_{j+1}^T$ in the vector of the optimal input sequence $\mathbf{U}$ is used while the remaining elements are discarded. The entire optimization procedure is repeated at the end of each rotation of the rotor. At the next rotation period $j+1$, the updated state $\bar{K}_{j+1}$ is used as an initial condition, while the cost function time limits in (23) roll ahead one step according to the receding horizon principle (Qin and Badgwell, 2003).

Equation (24) can be solved as a standard Quadratic Programming (QP) problem, by converting the control objectives in
(23) to the following form

$$J(\bar{K}_j, \mathbf{U}) = X^T \mathcal{Q} X + \mathbf{U}^T \mathcal{R} \mathbf{U}, \tag{26}$$

where $X = [\bar{K}_j, \bar{K}_{j+1}, \cdots, \bar{K}_{j+N_p}]^T$ corresponds to the vector of state predictions. $\mathcal{Q}$ and $\mathcal{R}$ are the weight matrices, which are

$$\mathcal{Q} = \operatorname{diag}(Q, \cdots, Q) \quad \mathcal{R} = \operatorname{diag}(R, \cdots, R), \tag{27}$$

By introducing the following prediction matrices,

$$\mathcal{A} = \begin{bmatrix} I \\ \bar{A}_j \\ \vdots \\ \bar{A}_j^{N_u} \\ \vdots \\ \bar{A}_j^{N_p} \end{bmatrix}, \qquad \mathcal{B} = \begin{bmatrix} 0 & \cdots & 0 \\ \hat{B}_j & \cdots & 0 \\ \vdots & \ddots & \vdots \\ \bar{A}_j^{N_u-1}\hat{B}_j & \cdots & \hat{B}_j \\ \vdots & \vdots & \vdots \\ \bar{A}_j^{N_p-1}\hat{B}_j & \cdots & \sum_{i=0}^{N_p-N_u} \bar{A}_j^i \hat{B}_j \end{bmatrix}, \tag{28}$$

equation (20) is rewritten as

$$X = \mathcal{A}\bar{K}_j + \mathcal{B}\mathbf{U}. \tag{29}$$

Combining (29) with (23), the MPC optimization can be solved in a simplified QP problem with only penalization of the
control input,

$$V(\bar{K}_j) = \min_{\mathbf{U}} (1/2) \cdot \mathbf{U}^T H \mathbf{U}, \tag{30}$$



subjected to

$$G \cdot \mathbf{U} \leq W, \tag{31}$$

where $H = 2\mathcal{R}$. In addition, $G$ and $W$ are defined according to (25), in a similar manner as in the paper (Bemporad et al.,
260    2002).

**Remark 1.** *The output constraints should be introduced in the control system cautiously as it could cause instability of the*
*predictive control system due to the non-linearity appearing in the control law and the model-plant mismatch (Wang, 2009).*
*To avoid this issue, we will introduce an output constraints relaxation, such that it allows the output constraints to be violated,*
*but incurring a heavy penalty cost. This is achieved by introducing the so-called slack variables $\rho$.*

As a result, the output constraints will be relaxed once the slack variables tend to large values during the control problem
formulation. The implementation of the slack variables in the objective function is

$$V(\bar{K}_j) = \min_{\mathbf{U}} (1/2) \cdot \mathbf{U}^T H \mathbf{U} + \rho^T F \rho, \tag{32}$$

subjected to

$$G \cdot \mathbf{U} - \rho \leq W, \tag{33}$$

where $F$ is the weight matrix for the slack variables. With the penalization of the slack variables, the output constraints are
softened to increase the control stability.

The implementation of the constrained repetitive control is schematically presented in Figure 2. As the Markov matrix $\hat{\bar{\Xi}}_k$
is derived from the online recursive subspace identification (8), the MPC optimization is implemented in (32)-(33). When the
blade loads of the wind turbine violate the output constraints, this cSPRC algorithm will formulate the repetitive control law for
load mitigation, as shown in (18). Otherwise, only the baseline controller is active to maintain the basic control performance,
thus leading to the reduced actuator activities.

## 4  Case study

In this section, the effectiveness of cSPRC in dealing with the output constraints is demonstrated on the wind turbine model
via a series of case studies. For the sake of comparisons, the load reduction and pitch ADC of the proposed cSPRC approach,
baseline CPC and MBC-based IPC are computed for investigations.

### 4.1  Model configuration

The wind turbine model, which has been introduced in Section 2, is simulated by the *FAST v8.16* simulation package (Jonkman
and Buhl, 2005). It is coupled with *Simulink* where the wind turbine torque and pitch control systems are implemented. Two
typical scenarios are considered in this paper, which are: (1) Wake-rotor overlap condition: the wind turbine is impinged by a





**Table 2.** Model configuration and environmental conditions in FAST-Simulink simulations for all LCs.

|  | LC 1-3: Wake overlapping case | LC 4-6: Turbulent wind case |
|---|---|---|
| Turbine | DTU 10MW | DTU 10MW |
| Inflow wind speed | 12 m/s, 16 m/s, 20 m/s | 12 m/s, 16 m/s, 20 m/s |
| Wind farm wake | FLORIS wake model (Gebraad et al., 2016) | - |
| Turbulence intensity | 0.00% | 3.75% |
| Simulation time | 1000 s | 1000 s |
| Time step | 0.01 s | 0.01s |

steady-state wind farm wake shed from an upstream turbine, which shows partial and full wake-rotor overlap. (2) Turbulent wind condition: the wind turbine is subjected to turbulent wind conditions.

The steady-state wind farm wake is simulated by the widely-used FLORIS model (Gebraad et al., 2016). In the parameterization of the FLORIS wake model, the Turbulence Intensity (TI) of $6.0\%$ and the center-to-center distance between the wake center and the downstream turbine rotor center (5 rotor diameters (5D)) are specified to characterize the wind farm wake. It implies that the simulated wind turbine in the FAST tool is situated 5D behind the upstream turbine. Other effects such as wake meandering, logarithmic wind profile and turbulence are not included in this scenarios. For the turbulent wind flow, a series of turbulent varying wind fields is simulated by using the TurbSim model[1], where the TI is set to be 3.75% while the inflow wind speeds are specified as 12m/s, 16m/s and 20m/s. These two scenarios result in a total of 6 Load Cases (LCs) in the case study. All of them are summarized in Table 2.

Then, the time series of the wind fields, which are based on the simulation results of the FLORIS and TurbSim, are fed into the FAST/Simulink model as the input of the wind turbine simulation. In all the LCs, the simulation lasts 1000s at a fixed discrete time step of 0.01s. Furthermore, the safety bounds for the blade loads, corresponding to the output constraints in cSPRC, are determined according to IEC 61400-1 regulation (IEC, 2005), which leads to

$$\bar{Y} = \xi \cdot \mathbb{Y}, \tag{34}$$

where $\xi$ is the safety factor and $\mathbb{Y}$ denotes the characteristic value for the loads, e.g. standard deviation of the loads. For the normal operating condition of the wind turbine, $\xi$ can be selected as 1.35 (IEC, 2005). For comparison, three different control strategies, namely the baseline CPC, MBC-based IPC and cSPRC, are simulated respectively in each LC. This finally leads to a total of $6 \cdot 3 = 18$ simulation runs.

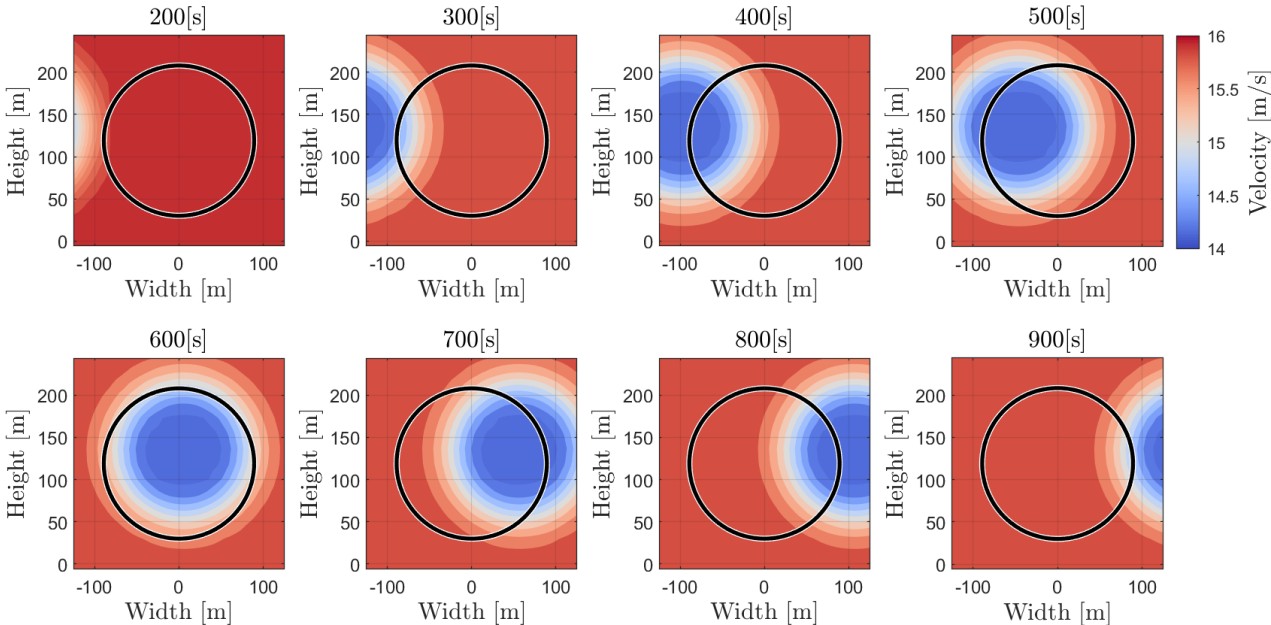

**Figure 3.** Vertical slice of the Wind field at the downstream turbine in LC2 (16m/s wind speed), where the rotor is impinged and overlapped by a wake. Red regions imply high wind velocity, which is undisturbed by the upstream turbine, while the blue regions implies a velocity deficit due to the upstream turbine.

## 4.2 Results and discussions

### 4.2.1 Scenario I: wake-rotor overlap

First of all, the propagation of the FLORIS wake to the downstream wind turbine is presented in Figure 3. As visible, the wind farm wake generated by the FLORIS model shows an in-wake velocity deficit, which is indicated by the blue regions in Figure 3. It will cause significant asymmetric loading on the rotor blades when the downstream turbine experiences the partial wake overlap (such as in 300s-500s and 700s-800s). The load control for such partial overlap induced asymmetric loads is demonstrated through a series of comparison studies. Figure 4 shows the time series of MOoP and corresponding pitch angles on one blade. In order to clearly demonstrate the performance of different IPC strategies, the steady values of MOoP, which are closely related to the pitch angles from the baseline CPC, have been removed in this figure.

It can be seen that MOoP is significantly increased when the wake impinges on the left sector of the rotor at around 350s, leading to the partial wake-rotor overlap. Due to the increase of MOoP, the proposed cSPRC actively generates the IPC action to reduce the asymmetric blade loads into the safety bounds and avoid violating the output constraints. Thus, significant load reduction can be observed during 350s-550s. As time goes by, the rotor is fully overlapped with the wake, which results in the reduced MOoP. Since the blade loads do not violate the output constraints at 600s, only the baseline CPC is active to

[1]TurbSim: a stochastic inflow turbulence tool to simulate realistic turbulent wind fields. https://nwtc.nrel.gov/TurbSim.





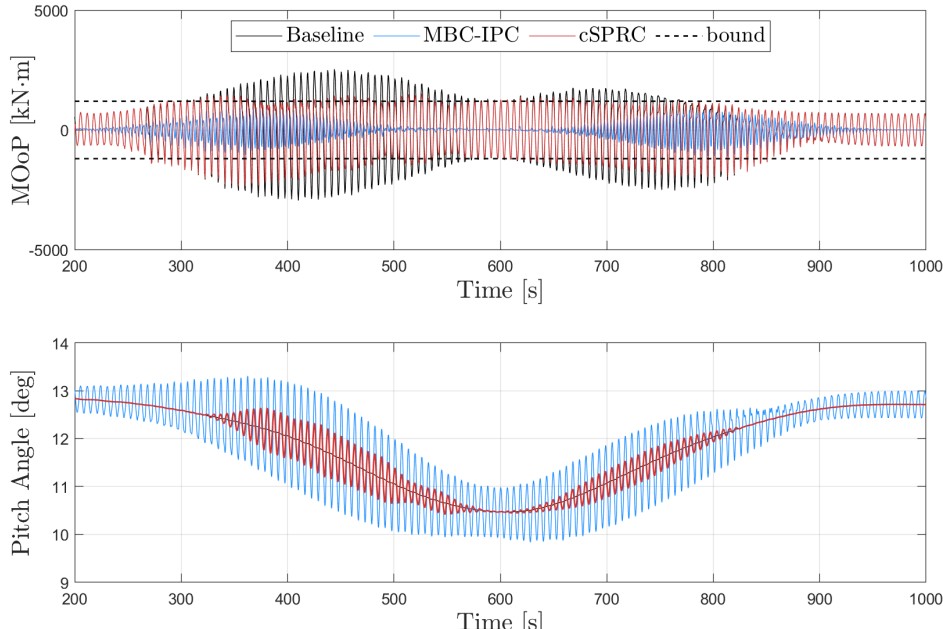

**Figure 4.** MOoP on the blade root in LC2 (16m/s wind speed) and its corresponding pitch angles, where the output constraints in the developed cSPRC are enabled at 200s. The steady-state values of MOoP have been removed. The designed safety bounds corresponding to the output constraints are 1200kN·m.

maintain the basic wind turbine performance, which leads to reduced actuator activities. Again, the wake impinged the right sector of the rotor at around 640s. The increased blade loads enables cSPRC to provide the IPC action for load mitigation. In comparisons, MBC-based IPC, which is a conventional IPC approach, actually shows maximum potential of load reduction. However, significant actuator activities are demanded by MPC-based IPC. For example, the corresponding pitch rates are presented in Figure 6. MBC-based IPC shows highest pitch rates, which lead to large cyclic fatigue loads on the actuators. The developed cSPRC, however, is only active in load mitigation when the blade loads violate the output constraints, thus significantly reducing the actuator activities.

The cost function of the MPC optimization in cSPRC is illustrated in Figure 5. It essentially implies the desired actuator activities for load mitigation in cSPRC. The value of the cost function increases when the wind turbine experiences the partial wake overlap. In the case where the blade loads satisfy the output constraints, the value of the cost function is reduced to zero. Note that as the slack variables are included in cSPRC, the output constraints are not always respected. Some violations can be still observed in 300s-400s in Figure 4. This will avoid the instability of the closed-loop control system due to the non-linearity occurring in the control law and the model-plant mismatch.

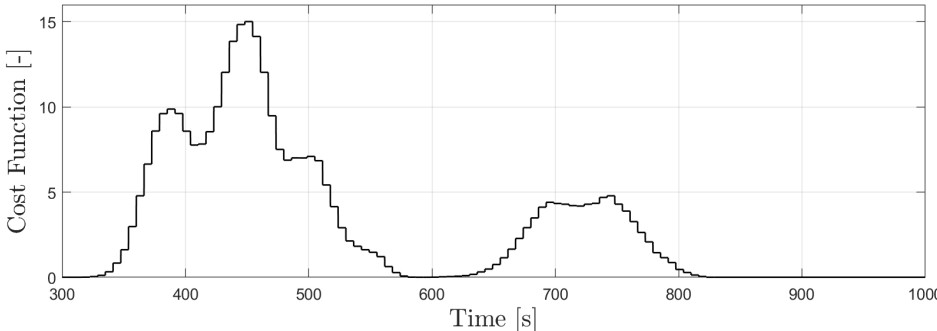

**Figure 5.** Cost function of the control objective in the developed cSPRC approach in LC2 (16m/s wind speed). The calculation of the cost function corresponds to (23) in Section 3.

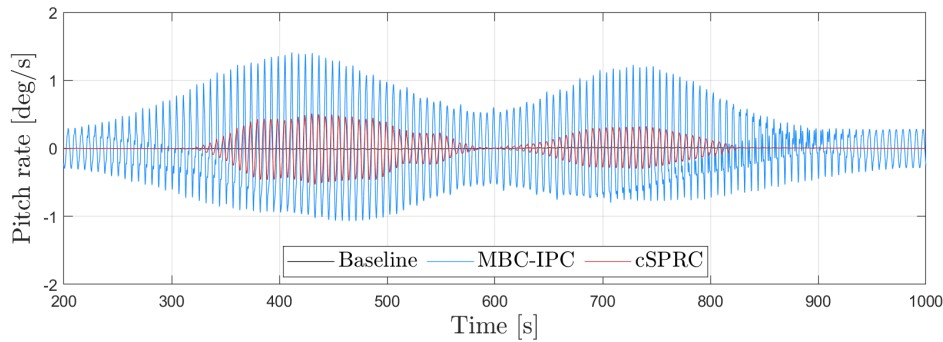

**Figure 6.** Pitch rate of the blade in LC2 (16m/s wind speed), where the output constraints in the developed cSPRC are enabled at 200s.

#### 4.2.2 Scenario II: Turbulent wind flow

Another scenario considered in the case study is the turbulent wind flow, where TI of $3.75\%$ is specified in this paper. Figs. 7-8 show the time series of MOoP, corresponding pitch angles and pitch rate in LC5. Similarly, the cSPRC formulates the IPC action for load reduction when the turbulence induced loads violate the output constraints. By using a tight safety bound of

335    500KN·m, we can see that significant load reduction is achieved by cSPRC while the output constraints are generally respected. Since the slack variables are used to relax the ouput constraints, some constraints violations are observed from Figure 7. On the other hand, MBC-based IPC shows maximum load reduction in the turbulent wind flow, however, it will induce more actuator activities, as indicated by the pitch rate in Figure 8. This approach, considering the blade loads are minimized into the safety bounds, is an effective way to reduce the actuator activities and deal with the output constraints.

340    Other LCs show similar patterns and hence are omitted for brevity. Based on these comparisons, it can be concluded that the developed cSPRC approach shows good performance in handling the output constraints in both wake overlap and turbulent wind flow scenarios. By designing safety bounds, it allows the wind farm operator to mitigate the loads into the safety bounds while reduce the actuator activities. However, the conventional approach, such as MBC-based IPC, usually mitigates the blade





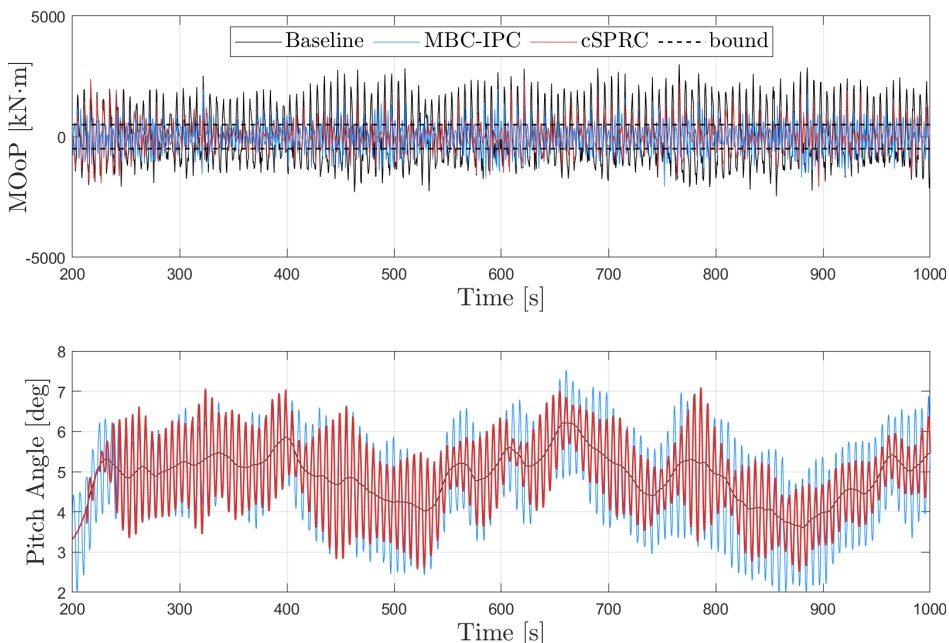

**Figure 7.** MOoP on the blade root and its corresponding pitch angles in LC4 (12m/s wind speed, TI 3.75 % case), where the output constraints in the developed cSPRC are enabled at 200s.The steady-state values of MOoP have been removed. The designed bound corresponding to the output constraints is 500kN·m.

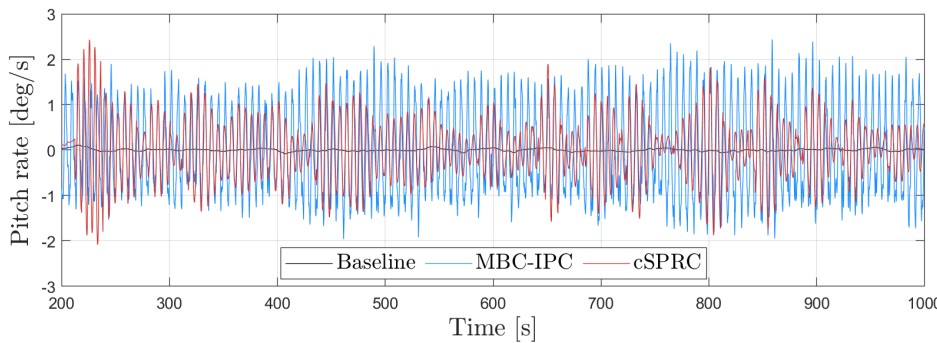

**Figure 8.** Pitch rate of the blade in LC4 (12m/s wind speed, TI 3.75 % case), where the output constraints in the developed cSPRC are enabled at 200s.

loads as much as possible. As a consequence, more actuator activities are demanded by the controller, which may lead to the
345    reduced reliability of the control system due to the higher cyclic fatigue loads on the pitch actuators.

In order to quantify the load reduction and pitch activities of these control strategies, two indicators, namely the the reduction of MOoP relative to baseline controller and the pitch ADC are calculated for comparisons. The latter one can be calculated





according to the pitch rate (Bottasso et al., 2013), which is defined as

$$\text{ADC} = \frac{1}{T} \int_0^T \frac{\dot{\beta}(t)}{\beta_{\max}} dt, \tag{35}$$

where $\dot{\beta}$ denotes the pitch rate while $\beta_{\max}$ is its maximum allowable value which is determined according to the specification of the wind turbine. $t$ is the time. For the 10MW wind turbine model, the value of $\beta_{\max}$ is

$$\beta_{\max} = \begin{cases} +8\text{deg/s}, & \dot{\beta}(t) \geq 0 \\ -8\text{deg/s}, & \dot{\beta}(t) < 0 \end{cases}, \tag{36}$$

Pitch ADC, which actually implies the cyclic fatigue loads on the pitch actuators, is a widely-used criterion to estimate the lifespan of pitch actuators. In addition, the ratio $r$ between the reduction of MOoP and pitch ADC is computed to comprehensively evaluate the control strategy. If $r$ is larger, the control strategy is more effective in load reduction with the same amount of pitch ADC, and vice versa. All these results are summarized in Table 3. In general, the proposed cSPRC shows similar or higher values of $r$ compared to MBC-based IPC, while its pitch ADC is significantly reduced in considered cases. For instance, cSPRC shows $\sim 1.7\%$ of pitch ADC in LC2, whereas MBC-based IPC shows $\sim 5.5\%$ of pitch ADC in this case. Averaging over all the cases, the proposed control strategy shows pitch ADC of $\sim 4\%$, thus leading to a higher $r$ of $8.5\%$. Considering the significant reduction of pitch ADC and higher $r$, it indicates that cSPRC is effective at reducing the actuator activities and maintain the same level of load mitigation. By incorporating the output constraints, this approach is able to minimize the loads into the designed safety bounds with low actuator activities.

In comparison, MBC-based IPC, aims at attaining the maximum load reduction. However, it causes excessive pitch ADC and thus leading to a lower $r$. According to Table 3, MBC-based IPC shows amount of pitch ADC with an average of $10.80\%$, which makes this approach not economical anymore. Therefore, it can be concluded that MBC-based IPC is not an economical way to deal with the blade loads due to the high actuator activities. Since cSPRC is able to take into account the output constraints, it would be only activated to provide the IPC actions for the necessary load reduction when the blade loads exceed the designed safety bounds. In this way, the proposed method is able to alleviate the pitch ADC, achieve more economical load mitigation and guarantee the safety of the wind turbine system in all considered cases.

## 5   Conclusions

In this paper, a fully data-driven Individual Pitch Control (IPC) approach, which is called constrained Subspace Predictive Repetitive Control (cSPRC), is developed to explicitly consider the output constraints in the control problem formulation. This approach involves using online recursive subspace identification and Model Predictive Control (MPC) to formulate the repetitive control law subjected to the output constraints. The cSPRC approach aims to produce the IPC action for load mitigation when the blade loads violate the output constraints while the baseline pitch controller is always active to maintain the basic wind turbine performance.



**Table 3.** Comparisons of the indicators (reduction of MOoP, pitch ADC and ratio $r$) in cSPRC and MBC-based IPC*.

|                        | LC1  | LC2  | LC3  | LC4  | LC5  | LC6  |
|------------------------|------|------|------|------|------|------|
| **Reduction of MOoP**  |      |      |      |      |      |      |
| cSPRC [%]              | 21.6 | 23.9 | 28.4 | 21.6 | 28.6 | 26.5 |
| MBC-IPC [%]            | 66.4 | 71.7 | 81.2 | 48.8 | 57.2 | 61.8 |
| **Pitch ADC**          |      |      |      |      |      |      |
| cSPRC [%]              | 6.2  | 1.7  | 1.5  | 4.3  | 5.8  | 6.5  |
| MBC-IPC [%]            | 18.8 | 5.5  | 4.7  | 11.3 | 11.5 | 13.0 |
| **Ratio $r$**          |      |      |      |      |      |      |
| cSPRC [-]              | 3.5  | 13.9 | 19.5 | 5.0  | 5.0  | 4.1  |
| MBC-IPC [-]            | 3.5  | 13.1 | 17.4 | 4.3  | 5.0  | 4.8  |

*The results are calculated based on the data from $300\,\mathrm{s} - 1000\,\mathrm{s}$ in wake overlap scenarios (LC1-LC3) and $750\,\mathrm{s} - 1000\,\mathrm{s}$ in turbulent wind flow scenarios (LC4-LC6). The reduction of MOoP represents the percentage of changes with respect to the baseline CPC case.

The effectiveness of the developed cSPRC in dealing with the output constraints is illustrated on a DTU 10MW reference wind turbine model, where the scenarios of wake-rotor overlap and turbulent wind flow are considered respectively. It proves that the cSPRC approach is effective at limiting the blade loads into the designed safety bounds, showing effective load mitigation with low pitch activities: the blade loads are reduced by $\sim 25\,\%$ while pitch Actuator Duty Cycle (ADC) is only $\sim 4\,\%$, thus leading to a ratio $r$ of $\sim 9\,\%$. Moreover, it is interesting to note that cSPRC only produces the IPC action for load mitigation when the blade loads violate the output constraints. This, to some extent, reduces the actuator activities.

In this paper, the cSPRC approach is compared to MBC-based IPC. The case study shows that MBC-based IPC attains maximum load reduction, however at the expense of increased pitch ADC. This may increase the wear on the bearings of the blades and reduce the reliability of the control system, which makes the conventional IPC approach not economical viable anymore. In comparison, the proposed cSPRC, by dealing with the output constraints, is capable of achieving more economical load reduction and shows much lower pitch ADC. More importantly, this approach enables the wind farm operator to design conservative bounds for the load control. Since the proposed cSPRC approach only formulates the IPC actions to prevent violating constraints, it will significantly alleviate the pitch ADC and extend the lifespan of the pitch control system. Future work will include, without being limited to, considering other wake effects such as wake meandering and dynamic propagation of the wake, executing scaled wind tunnel experiments, and full scale tests on a real wind turbine or wind farm.

*Video supplement.* This video shows the comparisons of load reduction between baseline control, MBC-based IPC and the proposed cSPRC strategies, where the wind turbine is impinged and overlapped by the wake shed from the upstream turbine. The ambient wind speed is 16m/s. The wind farm wake flow is simulated by the steady-state FLORIS model. The designed safety bounds in cSPRC are 1200kN· m.



*Acknowledgements.* This work was supported by the European Union via a Marie Sklodowska-Curie Action (Project EDOWE, grant 835901).



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
