# Peer review of "Load Reduction for Wind Turbines: an Output Constrained, Subspace Predictive Repetitive Control Approach"

_Wind Energy Science, 2021_

## Referee Comment (RC1)

**Review - Load Reduction for Wind Turbines: an Output Constrained, Subspace Predictive Repetitive Control Approach**

Torben Knudsen, Aalborg University

March 4, 2021

**Summary**

This paper is well written with good contributions. The methods and mathematics are very involved. On the one hand this is impressing but on the other it raises the question whether the same results could be achieved by simpler methods as also pointed out by Ervin Bossanyi. It could improve the paper if the authors elaborated on this issue. Another question is what role and impact the adaptivity in the data driven modeling has for the results. I recommend publication after corrections and revision.

**Specific comments**

1. P1 In the introduction the IPC approach is very much motivated by application in wind farms. As discussed in e.g. Knudsen et al. [2015] some of the promising wake control methods are active yawing or IPC for wake redirection. Please explain how your solution fits into this kind of farm control.

2. Figure 2. This figure is not so clear when seen initially. What does the coloring mean. What does the gray block with $U_f^+$ mean etc?

3. P7

   (a) "where $P$ is the period of the disturbance, equalling to the blade rotation period" Then the sampling time must be an integer of $P$ which is changing all the time! How do you solve that?

   (b) The paper says "Then the effect of the periodic blade loads $d_k$ on the input-output system can be eliminated as it holds" What does "as it holds" mean? This must be due to assumptions or definitions? Please explain?

4. P8

   (a) In (8) I guess it should be $\lambda^{k-i}$ ?

   (b) Please explain how the "persistent exciting signals" are chosen?

5. P11 Line 215 Why is the dimension of $\bar{A}$ $12l$? Maybe it is so but it is not clear to me.

6. P14 TI for LC 1-3 is stated as 0 in table but 6 in text!

**References**

T. Knudsen, T. Bak, and M. Svenstrup. Survey of wind farm control - power and fatigue optimization. *Wind Energy*, 18(8):1333–1351, August 2015. doi: 10.1002/we.1760. Published online 9 May 2014 in Wiley Online Library (onlinelibrary.wiley.com).

---

## Referee Comment (RC2)

**Review of the paper entitled: "Load Reduction for Wind Turbines: an Output Constrained, Subspace Predictive Repetitive Control Approach", by Liu Y. et al.**

**General comment:**

The paper deals with a novel control strategy aimed at reducing the oscillating loads on rotor blades.

The algorithm is based on a predictive control implementing constraints on a particular output. The formulation is based on techniques already present in literature (receding horizon control, recursive identification, constrained controls), that are here used in synergy for generating a particular control action.

The constraint is used to trigger the control when a selected output (here out-of-plane moment) exceed a selected threshold. In the end, the control is able to reduce (not cancel) the loads only when it is necessary, i.e., when the loading is too high. Because of that, the impact of the control on actuator duty cycle is extremely limited.

The performance of the proposed control is compared with that of a widespread technique aimed similarly at reducing oscillating loads, that is the standard IPC based on the Coleman transformation. In this regard, I have to say that the two controllers have similar but substantially different scopes. Standard IPCs can target and remove completely the oscillating loads thanks to the integral feedback. To do so, the use of the actuators results high. The proposed control, on the other side, acts so as to "simply" limit the oscillations under a desired threshold. This latter represents an extremely interesting control philosophy, which may find applicability outside the goal of the present work.

Although the comparison performed in the work (cSPRC vs MBC-IPC) is clearly important, I hope that the Authors in an amended version of the manuscript may acknowledge that the scope of the two controllers is "similar but different", and that, eventually, cSPRC is not the "winner" but rather a good alternative to MBC-IPC with specific pros and cons.

The paper is well-written and represents a good piece of research. I recommend the acceptance of the paper with minor corrections. The suggestions are all listed here in the following. The list in "Minor comments" refers to corrections that can be regarded as "minor" but worthy of a particular attention.

**Minor comments:**

1. RLS problem requires a persistent excitation (as also reported by Authors, see lines 168-169). This point deserves additional explanation. I may pose some questions to emphasize the issues which could be connected to that:
   a. How is this excitation practically imposed? Pitch actuation? Atmospheric turbulence?
      i. If the persistent excitation is achieved by pitch, this will have a great impact on ADC, which will be detrimental in light of the main aim of the work (see line 8 "actuator activities can be significantly reduced"). Moreover, what is here the role of the actuator bandwidth? Is it possible to have a pitch-driven persistent excitation within the frequency band of interest?
      ii. In case of excitation provided by turbulence, can it be considered "persistent" in the frequency band of interest? What one can say for very low turbulence inflow typical of offshore winds?
   b. Since the system is periodic and made by three rotating blades, the requirement of persistent excitation is to be carefully considered. One cannot simply check the excitation

made by a collective pitch motion, but also in the other two component (sin and cos, see the Coleman transformation).

      i. If my doubt is well-founded, this can have an important impact on loads, as the rotor will be always subject to an excitation which could modify nodding and yawing rotor moments.

2. Line 230: the whole methodology is attractive as it is fully automated and adaptive. Apparently, it does not need a dedicated tuning campaign, as it can be expected for standard PID controller. Unfortunately, once the constraint is enforced in the problem, one needs to minimize a cost function depending on two user-defined matrices $Q$ and $R$. Some indications for the selection of these matrices are important. Consider also that the number of entries in both $Q$ and $R$ can be high even if one assumes them diagonal.

3. Section 4.1: Are the pitch actuator dynamics considered in the model? If not, is it possible to infer how much the presence of actuator dynamics may affect the outputs for all analyzed controls, especially at the multiples of the rotor frequency (if considered in the controllers)?

4. All in all, I may imagine that one may force MBC-IPC (even if integral feedback is present) to have limited ADC, through the imposition of upper boundaries in the cosine and sine control variables (the signals before the anti-Coleman transformation) and through suitable anti-windup filters. If Authors share my opinion, they could note it somewhere in the text (e.g., introduction).

**Technical corrections:**

1. Abstract. I would smooth the sentence "… the current IPC design not economically viable". It is hard to demonstrate that IPC, considering the entire turbine, is not economically viable.

2. Line 55: change "to the best of author's knowledge" in "to the best of authors' knowledge".

3. Line 62: There is the repetition of the author's name inside and outside the parenthesis. Probably using \citet in latex may give a better output.

4. Figure 2: I believe that the arrow between the "10 MW Wind Turbine (Plant)" and the "Basis functions" is there to emphasize the fact that the sinusoidal are based on the actual azimuth angle coming from the plant. If so, the symbol $\varphi$ can be associated to the arrow itself.

5. Line 144: change one $\delta u$ in $\delta x$

6. Line 224: The meaning of $\bar{T}$ is not defined. Could it be $\bar{Y}$? Moreover, in many places it is written that the constraint is defined according to the IEC standards. Those sentences seem a bit vague. How is it practically defined?

7. Section 4.2.1: During the simulation, does the wake travel from left to right as it can be inferred by Figure 3? If so, this should be specified in the text to ease the comprehension.

8. Figure 4: could it be interesting to show also in this case the yawing moment?

9. Figure 4: the caption reads "The steady-state values of MOoP have been removed". This is ok for clarity of the picture, but, at the same time, I may say that in a real environment this cannot be done and, in turn, one has to define variable bounds. In fact, the limitation should be different for different wind speed (or even TI). This should be noted during the mathematical treatment as it could represent an additional complexity in the control scheme.

10. Section 4.2.2: It is not clear whether within the turbulent flow also non-null shear layer is considered.

11. Conclusion: I would expect a comment on multifrequency IPC. Is it possible to easily extend the cSPRC algorithm to multifrequency IPC? Such an extension may be useful for reducing loads in the fixed system, for example the 3p loads at hub, which can be mitigated through 2p and/or 4p IPC.

---

## Author Response (AR1)

| | |
|---:|:---|
| Date | November 25, 2021 |
| Our reference | n/a |
| Your reference | Manuscript WES-2021-2 |
| Contact person | Yichao Liu |
| Telephone/fax | +31 (0)64 77 39881 / n/a |
| E-mail | y.liu-17@tudelft.nl |
| Subject | Author's Response |

**Delft University of Technology**

Delft Center for Systems and Control

Address
Mekelweg 2 (3ME building)
2628 CD Delft
The Netherlands

Editor and reviewers
*From Wind Energy Science*

Dear editor and reviewers,

The authors would like to thank you for the positive comments and constructive suggestions to our paper. We believe that your feedback has helped us to significantly improve the quality of the manuscript.

As required, a careful revision has been made to the paper, in order to take into account all the feedback. The objective of this document is to reply to the points raised in the reviews and to provide a detailed overview of the changes made. For each comment, a point-to-point response is provided in blue color, while the corresponding changes to the manuscript are reported in red. Moreover, we have prepared two versions of manuscripts. One of them highlights all the revised portions in red. The other one is the clean version for your reference.

Yours sincerely,

Yichao Liu
on behalf of the other authors

Enclosure(s): Response to comments of Dr. Torben Knudsen
Response to comments of Reviewer 2

**Response to comments of Dr. Torben Knudsen**

**Summary**

This paper is well written with good contributions. The methods and mathematics are very involved. On the one hand this is impressing but on the other it raises the question whether the same results could be achieved by simpler methods as also pointed out by Ervin Bossanyi. It could improve the paper if the authors elaborated on this issue. Another question is what role and impact the adaptivity in the data driven modeling has for the results. I recommend publication after corrections and revision.

**Response**:
First of all, we would like to thank the reviewer for the positive feedback. Second, we will reply to all the points raised in your review and provide a detailed overview of the revisions we made. The corresponding answers to the questions raised in Summary are here:

(a) Over the past years, a large number of individual pitch control algorithms have been developed to achieve the load reduction, such as proportional integral control, model predictive control, etc. With a traditional PI-based IPC, augmented with some deadbands, we might be able to achieve something similar. However, we haven't found something similar in literature, taking into account the output constraints in load control and achieve the same results. Moreover, the selection of those parameters can be a potential issue as there is no mathematically rigorous way to tune its parameters.
In addition, there exist a couple of other advantages of the proposed method: 1) can directly minimize a cost function, 2) is predictive 3) can be used directly in a MIMO setting (no need for decoupling) 4) has more degrees of freedom than a PID controller (depending on the windows used) 5) can directly be made adaptive without increasing the complexity.
In the revised version, this issue has been elaborated in Section 1 (Introduction) and 5 (Conclusions) in order to improve the paper and hopefully address the reviewer's concerns.

(b) The role and impact of the adaptivity in the proposed data driven method are clarified as follows. The wind turbine operating in different operating conditions shows nonlinear dynamics and uncertainties. Thanks to the adaptivity from the online recursive system identification, the proposed method can be adapted to varying operating conditions automatically by learning and updating the wind turbine dynamics online.

Therefore, the main role of the adaptivity is to keep the internal wind turbine model of the cSPRC adapted to the varying operating conditions by learning the wind turbine dynamics online. This adaptivity introduces positive impact on this data-driven method, as it will help cope with the challenges from the nonlinear wind turbine dynamics and the wind uncertainties.

In the revised version, the role and impact of the adaptivity has been explicitly indicated in Section 3.1.

Furthermore, a couple of revisions have been made to improve the quality of the manuscript according to the reviewer's specific comments, which are summarized in this letter. We hope the revisions have sufficiently addressed the reviewer's concerns.

**Revised portion**:
(Page: 2, Line: 57-59)
The widely-used Proportional Integral (PI)-based IPC augmented with some deadbands, might be able to take into account the output constraints, while the tuning procedure of the parameters is rather cumbersome.

(Page: 8, Line: 174-176)
Thanks to the online recursive system identification, $\hat{\Xi}_k$ will be adapted to different operating conditions by learning the wind turbine dynamics online. Then it is used in an MPC framework to formulate a repetitive control law subjected to the output constraints.

**Specific comments**

1. P1 In the introduction the IPC approach is very much motivated by application in wind farms. As discussed in e.g. Knudsen et al. [2015] some of the promising wake control methods are active yawing or IPC for wake redirection. Please explain how your solution fits into this kind of farm control.

**Response**:
We agree with the reviewer that the IPC is a promising approach for wind farm wake control. In a wind farm, the wake will lead to increased fatigue loads and loss of power. The objective of wind farm strategies is to maximize the total wind farm power, track the power reference, and minimize the fatigue loading on the wind farm, which is matching the main goal of the proposed our cSPRC solution. The proposed algorithm is a fully data-driven method, which can be extended to a wind farm level to achieve the wake load control. The wake induced fatigue loads can be significantly mitigated by the pitch control law. A Similar concept of the IPC for the wind farm wake control can be found in reference [1].

[1] Frederik, JA, Doekemeijer, BM, Mulders, SP, van Wingerden, J. W. The helix approach: using dynamic individual pitch control to enhance wake mixing in wind farms. Wind Energy. 2020; 23: 1739– 1751. https://doi.org/10.1002/we.2513

Furthermore, the output constraints will guarantee that only the blade loads violating the designed bounds will be mitigated. This function will fit into the farm control, as the wake-turbine overlap, as an example illustrated in our paper, commonly occurs in a wind farm. The proposed cSPRC solution will significantly reduce the pitch actuator activities and still maintain critical load reduction.

In the revised paper, the literature provided by the reviewer has been included in the introduction for reference.

**Revised portion**:
(Page: 2, Line: 39-40)
On the other hand, the application of IPC to wake load control in a wind farm is receiving increasing attention (Knudsen et al., 2015).

2. Figure 2. This figure is not so clear when seen initially. What does the coloring mean. What does the gray block with $U_f^+$ mean etc?

**Response**:
In the revised paper, Figure 2 has been improved. In detail, the coloring has been removed, and all the symbols and letters in the figure have been improved. In addition, the meaning of the symbols are clearly indicated in the caption. The updated Figure 2 is also attached here for the reviewer's convenience.

**Revised portion**:
(Page: 6, Figure: 2)

[Figure]

Figure 2. Implementation of cSPRC, which includes online system identification and repetitive control. MPC optimization is used to incorporate the output constraints in repetitive control formulation. $U_f$ represents the basis function, while the symbol $U_f^+$ denotes the Moore-Penrose pseudo-inverse of the basis function. In addition, $u_k$, $y_k$ are the input and output vectors at discrete time index $k$. $\theta_j$, $\tilde{Y}_j$, and $\bar{Y}_j$ denote the transformed control input, transformed control output and the output constraints at rotation $j$. $P$ and $\varphi$ correspond to the period of the disturbance and rotor azimuth.

3. P7 (a) "where P is the period of the disturbance, equalling to the blade rotation period" Then the sampling time must be an integer of P which is changing all the time! How do you solve that?

(b) The paper says "Then the effect of the periodic blade loads dk on the input- output system can be eliminated as it holds" What does "as it holds" mean? This must be due to assumptions or definitions? Please explain?

**Response**:
(a) The blade rotation period is indeed time varying, due to the wind turbulence and the change in the inflow wind speed. It is also true that the discrete time period $P$ is assumed to be such that $P\Delta t$, where $\Delta t$ here is the sampling time, should be equal to the rotation period in seconds. This issue is solved via the following two steps:

i. In the implementation as shown in Figure 1 (Page 5), we have a baseline feedback PI pitch controller to guarantee that in the above-rated wind condition, the wind turbine operates at the rated operating point with a specific rated rotation period of 6.25s. On the other hand, the sampling time $\Delta t$ is a constant and small value, i.e., 0.07s in our implementation. This will lead to a good approximation of the rotation period, and the difference between $P\Delta t$ and the steady-state rated rotation period can be neglected.

ii. The potential variation of the rotor speed.
Still, one issue should be handled in this approach is the potential variation of rotor speed, due to the wind turbulence or the changes in the inflow wind speed. This will lead to a phase shift between input and output. To deal with this issue, the rotor azimuth, i.e., $\psi_k$ measured through the shaft encoder in this paper, which equals the value of $2\pi k/P$ at time instant $k$ is used in the basis function, instead of the fixed values ($2\pi/P$, $4\pi/P$, etc.) in equation (17) is utilized in the basis function. The will guarantee the proposed cSPRC algorithm is able to take into account the potential variations in rotor speed.

(b) 'it holds' in this sentence means that the following equation (1) is satisfied, which will lead to the elimination of the periodic blade loads $d_k$:

$$\delta d_k = d_k - d_{k-P} = 0\,. \tag{1}$$

In order to make it clearer, the relevant narrative has been improved, and additional remark has been included in the revised paper.

**Revised portion**:
(Page: 10, Line: 208-212)
**Remark 1.** One issue need to be addressed in the cSPRC algorithm is the potential variation of rotor speed due to the varying inflow wind speed. This will result in a phase shift between control input and output. To solve this problem, the rotor azimuth $\psi$, equal to $2\pi k/P$ at time instant $k$, is utilized to reformulate $U_f$ to take into account the rotor speed variations. In this context, equation (17) can be rewritten into the form:

$$\phi = \underbrace{\left[\; \sin(\psi) \quad \cos(\psi) \;\right]}_{U_f} \otimes I_r\,, \tag{2}$$

(Page: 7, Line: 146)
...as the following equation holds: ...

4. P8 (a) In (8) I guess it should be $\lambda^{k-i}$?

(b) Please explain how the "persistent exciting signals" are chosen?

**Response**:

(a) $\lambda$ is a forgetting factor to alleviate the past data in this algorithm. In our case, it is a predefined constant parameter. So we use $\lambda$ here without any superscript.

(b) For the persistent exciting signals, we selected the filtered pseudo-random binary noise, which is deemed as an ideal exciting signal for system identification [2], to excite the wind turbine system dynamics.
More specific, the exciting signals are superimposed on top of the transformed control input $\theta_j$ as
$$U_k^{(P)} = \phi(\theta_j + \eta_j)\,,$$
where the vector $\eta_j \in \mathbb{R}^{b \cdot r}$ is the filtered pseudo-random binary noise, which is used to excite system at the 1P frequency. Because of the transformation matrix $\phi$ in equation $(5)$, the energy of the persistently exciting control input $U_k^{(P)}$ is projected onto the specified 1P frequency. In order to guarantee the successful excitation, $\eta$ should be generated in an uncorrelated way with different random seeds for each component of the vector $\theta$.

[2] Tulleken, H., Generalized binary noise test-signal concept for improved identification-experiment design, Automatica, 1990; 26: 37-49.

The explanation of the chosen persistent exciting signals has been added to the revised paper for clarifications.

**Revised portion**:
(Page: 11, Line: 217-223)
To excite the wind turbine system dynamics, the persistent exciting signals are super-imposed on top of the transformed control input $\theta_j$. The control inputs now are given by
$$U_k^{(P)} = \phi \cdot (\theta_j + \eta_j)\,, \tag{3}$$

where the vector $\eta_j \in \mathbb{R}^{b \cdot r}$ is the filtered pseudo-random binary noise. Thanks to the transformation matrix $\phi$, the energy of the persistently exciting control input $U_k^{(P)}$ can be restricted on the specified 1P frequency as well. This will alleviate the negative effects of the excitation on the nominal wind turbine dynamics. Furthermore, $\eta$ is generated in an uncorrelated way with different random seeds for each component of the vector $\theta$ to guarantee the successful excitation.

5. P11 Line 215 Why is the dimension of $\bar{A}$ 12$l$? Maybe it is so but it is not clear to me.

**Response**:
We would like to thank the reviewer for pointing out this error. Indeed, the dimension of the matrix $\bar{A}$ is not $12l$. The row of this matrix should be $(2lb + 2rb)$, and the column of this matrix is $4rb$.
Furthermore, the dimension of all the matrices has been double checked and corrected in the revised paper.

**Revised portion**:
(Page: 11, Line: 229-231)
Compared to (16), the dimension of the projected matrix, *i.e.*, $\bar{A} \in \mathbb{R}^{2(lb+rb) \times 4rb}$, is much lower than the original matrix $\hat{A} \in \mathbb{R}^{2(lP+rP) \times 4rP}$. Considering $P \gg b$, the order of the state-space representation as well as the following MPC optimization problem can be substantially reduced.

(Page: 11, Line: 216)
$\theta \in \mathbb{R}^{b \cdot r}$,...

(Page: 11, equation (22))

$$\underbrace{\begin{bmatrix} \tilde{Y}_{j+1} \\ \delta\theta_{j+1} \\ \delta\tilde{Y}_{j+1} \\ \theta_{j+1} \end{bmatrix}}_{\bar{\mathcal{K}}_{j+1}} = \underbrace{\begin{bmatrix} I_{l \cdot b} & \phi^+ \Gamma^{(P)} \widehat{K_u^{(P)}} \phi & \phi^+ \Gamma^{(P)} \widehat{K_y^{(P)}} \phi & -\phi^+ \hat{H}^{(P)} \phi \\ 0_{r \cdot b \times l \cdot b} & 0_{r \cdot b} & 0_{r \cdot b \times l \cdot b} & -I_{r \cdot b} \\ 0_{l \cdot b} & \phi^+ \Gamma^{(P)} \widehat{K_u^{(P)}} \phi & \phi^+ \Gamma^{(P)} \widehat{K_y^{(P)}} \phi & -\phi^+ \hat{H}^{(P)} \phi \\ 0_{r \cdot b \times l \cdot b} & 0_{r \cdot b} & 0_{r \cdot b \times l \cdot b} & 0_{r \cdot b} \end{bmatrix}}_{\bar{A}_j} \underbrace{\begin{bmatrix} \tilde{Y}_j \\ \delta\theta_j \\ \delta Y_j \\ \theta_j \end{bmatrix}}_{\bar{\mathcal{K}}_j} + $$

$$\underbrace{\begin{bmatrix} \phi^+ \hat{H}^{(P)} \phi \\ I_{r \cdot b} \\ \phi^+ \hat{H}^{(P)} \phi \\ I_{r \cdot b} \end{bmatrix}}_{\hat{B}_j} \theta_{j+1} .$$

6. P14 TI for LC 1-3 is started as 0 in table but 6 in text!

**Response**:
We thank the reviewer for pointing out this misleading information. In the FLORIS wake model, the turbulence intensity is one of the input parameters in FLORIS, which is only utilized to define the wake recovery of the simulated FLORIS wake [3,4]. However, there is no turbulence involved in the FLORIS wake model.

[3] FLORIS 2021, FLORIS. Version 2.4, GitHub repository.
https://github.com/NREL/floris

[4] Doekemeijer, B., van der Hoek, D., van Wingerden J. W., Closed-loop model-based wind farm control using FLORIS under time-varying inflow conditions. Renewable Energy, 156: 719-730, 2020.

The relevant narrative has been revised and improved in order to make it clear.

**Revised portion**:
(Page: 14, Line: 309-313)
In the parameterization of the FLORIS wake model, the Turbulence Intensity (TI) of $6.0\%$ is utilized to define the wake recovery, while the center-to-center distance between the wake center and the downstream turbine rotor center (5 Diameters (5D) of the rotor) is specified, which implies that the simulated wind turbine in the FAST tool is situated 5D behind the upstream turbine. Other effects such as wake meandering, turbulence logarithmic wind profile are not included in this scenario.

(Page: 15, Table 2)
The value of the turbulence intensity for LC1-3 in this Table has been removed.

**Response to comments of Reviewer 2**

**General comment:**

The paper deals with a novel control strategy aimed at reducing the oscillating loads on rotor blades.

The algorithm is based on a predictive control implementing constraints on a particular output. The formulation is based on techniques already present in literature (receding horizon control, recursive identification, constrained controls), that are here used in synergy for generating a particular control action.

The constraint is used to trigger the control when a selected output (here out-of-plane moment) exceed a selected threshold. In the end, the control is able to reduce (not cancel) the loads only when it is necessary, i.e., when the loading is too high. Because of that, the impact of the control on actuator duty cycle is extremely limited.

The performance of the proposed control is compared with that of a widespread technique aimed similarly at reducing oscillating loads, that is the standard IPC based on the Coleman transformation. In this regard, I have to say that the two controllers have similar but substantially different scopes. Standard IPCs can target and remove completely the oscillating loads thanks to the integral feedback. To do so, the use of the actuators results high. The proposed control, on the other side, acts so as to "simply" limit the oscillations under a desired threshold. This latter represents an extremely interesting control philosophy, which may find applicability outside the goal of the present work.

Although the comparison performed in the work (cSPRC vs MBC-IPC) is clearly important, I hope that the Authors in an amended version of the manuscript may acknowledge that the scope of the two controllers is "similar but different", and that, eventually, cSPRC is not the "winner" but rather a good alternative to MBC-IPC with specific pros and cons.

The paper is well-written and represents a good piece of research. I recommend the acceptance of the paper with minor corrections. The suggestions are all listed here in the following. The list in "Minor comments" refers to corrections that can be regarded as "minor" but worthy of a particular attention.

**Response**:

First, we would like to thank the reviewer for the positive comments.

Second, we agree with the reviewer that the proposed cSPRC framework and the standard IPC based on the Coleman transformation have similar but substantially different scopes. We acknowledge that the scope of the two controllers is "similar but different" and they can be used in different application scenarios with different control objectives.

Third, the "similar but different" scopes of these two controllers have been indicated in the amended version of this manuscript, according to the reviewer's suggestion. In addition, a careful revision has been made to improve the paper based on the reviewer's other comments. The point-by-point response to the reviewer's comments are as below.

**Revised portion**:

(Page: 20, Line: 383-386)

Since the output constraints are taken into account in cSPRC, the proposed framework is able to only mitigate the blade loads violating the designed safety bounds. In this way, the pitch ADC is significantly alleviated. Therefore, cSPRC might be a promising alternative to MBC-based IPC to perform a trade-off between load reduction and pitch ADC in all considered cases.

(Page: 21, Line: 406-409)

Based on the comparison study, it is worth noting that both cSPRC and MBC-based IPC show similar but substantially different scopes. MBC-based IPC targets a maximum load reduction at the expense of high pitch ADC. cSPRC might be a complementary alternative to MBC-based IPC to achieve a trade-off between load reduction and pitch ADC.

**Minor comments:**

1. RLS problem requires a persistent excitation (as also reported by Authors, see lines 168-169). This point deserves additional explanation. I may pose some questions to emphasize the issues which could be connected to that:

    a. How is this excitation practically imposed? Pitch actuation? Atmospheric turbulence?

        i. If the persistent excitation is achieved by pitch, this will have a great impact on ADC, which will be detrimental in light of the main aim of the work (see line 8 "actuator activities can be significantly reduced"). Moreover, what is here the role of the actuator bandwidth? Is it possible to have a pitch-driven persistent excitation within the frequency band of interest?

        ii. In case of excitation provided by turbulence, can it be considered "persistent" in the frequency band of interest? What one can say for very low turbulence inflow typical of offshore winds?

b. Since the system is periodic and made by three rotating blades, the requirement of persistent excitation is to be carefully considered. One cannot simply check the excitation made by a collective pitch motion, but also in the other two component (sin and cos, see the Coleman transformation).

    i. If my doubt is well-founded, this can have an important impact on loads, as the rotor will be always subject to an excitation which could modify nodding and yawing rotor moments.

**Response**:

First of all, we would like to thank the reviewer for the critical and helpful comment, which helps us improving the paper. Indeed a persistent excitation, which is widely used to solve the RLS problem, is utilized in this algorithm. Below are our answers to the reviewer's two questions in this comment:

(a) **The excitation is imposed by pitching the individual blade without significant effects on ADC.**

In order to sufficiently answer the reviewer's question, the persistent excitation method is detailed as follows:

First, we chose the filtered pseudo-random binary noise as the persistent excitation in this study. Usually it is considered as an ideal exciting signal for the system identification [1].

Second, the persistent exciting signals are superimposed on top of the transformed control input $\theta_j$. The control inputs now are given by

$$U_k^{(P)} = \phi \cdot (\theta_j + \eta_j)\,, \tag{4}$$

where the vector $\eta_j \in \mathbb{R}^{b \cdot r}$ is the filtered pseudo-random binary noise, which is only used to excite the system at the 1P frequency. Thanks to the transformation matrix $\phi$ in equation (5), the energy of the persistently exciting control input $U_k^{(P)}$ is restricted on the specified 1P frequency as well with limited effects on ADC. Therefore, the bandwith of the pitch-driven persistent excitation is within the frequency band of interest, *i.e.*, 1P frequency, indeed.

(b) **The persistent excitation is superimposed on top of the transformed individual pitch control inputs $\theta$ as individual pitch motions.**

This consideration is based on the fact that the wind turbine system is periodic and made by three rotating blade as the reviewer pointed out. Therefore, the persistent exciting noise on top of the individual control inputs would consist of sin and cos components. In addition, the noise $\eta_j$ in above-mentioned equation (1) should be generated in an uncorrelated way with different random seeds for each component of the vector $\theta$. The bandwith of the noise is limited to the frequency of the interests because of the transformation matrix $\phi$. Therefore, the excitation will have very limited effect on the rotor.

[1] Tulleken, H., Generalized binary noise test-signal concept for improved identification-experiment design, Automatica, 1990; 26: 37-49.

Based on this comment, the persistent excitation is elaborated in the revised paper for clarifications.

**Revised portion**:
(Page: 11, Line: 217-223)
To excite the wind turbine system dynamics, the persistent exciting signals are super-imposed on top of the transformed control input $\theta_j$. The control inputs now are given by

$$U_k^{(P)} = \phi \cdot (\theta_j + \eta_j), \tag{5}$$

where the vector $\eta_j \in \mathbb{R}^{b \cdot r}$ is the filtered pseudo-random binary noise. Thanks to the transformation matrix $\phi$, the energy of the persistently exciting control input $U_k^{(P)}$ can be restricted on the specified 1P frequency as well. This will alleviate the negative effects of the excitation on the nominal wind turbine dynamics. Furthermore, $\eta$ is generated in an uncorrelated way with different random seeds for each component of the vector $\theta$ to guarantee the successful excitation.

2. Line 230: the whole methodology is attractive as it is fully automated and adaptive. Apparently, it does not need a dedicated tuning campaign, as it can be expected for standard PID controller. Unfortunately, once the constraint is enforced in the problem, one needs to minimize a cost function depending on two user-defined matrices $Q$ and $R$. Some indications for the selection of these matrices are important. Consider also that the number of entries in both $Q$ and $R$ can be high even if one assumes them diagonal.

**Response**:
We would like to thank the reviewer for this positive comment. Indeed the proposed cSPRC method is made automated and adaptive, and weighting matrices are usually defined to perform the trade-off between the stability and the convergence rate. For the tuning procedure of the weighting matrices, we would like to clarify that:

(a) The matrix $Q$, corresponding to the penalization of the states, is set to an all-zeros weighting matrix, as the controller is designed to be only active when the blade loads violate the output constraints. Therefore, there is no need to tune the matrix $Q$ in this algorithm. In the revised paper, we have simplified relevant mathematics and clarified $Q$ is an all-zeros weighting matrix.

(b) The selection of the diagonal matrix $R$ is important indeed as pointed out by the reviewer. The element on the diagonal locations of $R$ corresponds to the penalization of each element of the control input vector. As the controller should be able to respond to the output violation in a short time, a small value is selected for each element of $R$ to guarantee a quick response to the output violations and also perform a good trade-off between the control stability and the convergence rate. In the simulation case, all the element of the diagonal matrix $R$ is set as $1$.

As we only need to tune the weighing matrix $R$, above-mentioned tuning procedure would be less cumbersome. More indications for the selection of these matrices have been included in the revised version.

**Revised portion**:
(Page: 12, Line: 257-260)
Since the cSPRC framework is designed to be only active when the blade loads violate the output constraints, $Q$ is set to an all-zeros weighting matrix. This will make $\theta_{j+i|j}$ the only penalization term in the cost function. To guarantee a quick response to the output violations and also perform a good trade-off between the control stability and the convergence rate, a value close to $1$ is selected for each element of $R$.

(Page: 13, Line: 267-271)
With the all-zeros weighting matrix $Q$ in mind, the control objectives in (27) can be converted to the following form

$$J(\bar{\mathcal{K}}_j, \mathbf{U}) = \mathbf{U}^T \mathcal{R} \mathbf{U},\tag{6}$$

where $X = [\bar{\mathcal{K}}_j, \bar{\mathcal{K}}_{j+1}, \cdots, \bar{\mathcal{K}}_{j+N_p}]^T$ corresponds to the vector of state predictions. $\mathcal{R}$ is the weight matrix, which is

$$\mathcal{R} = \begin{bmatrix} R & & \\ & \ddots & \\ & & R \end{bmatrix}.\tag{7}$$

3. Section 4.1: Are the pitch actuator dynamics considered in the model? If not, is it possible to infer how much the presence of actuator dynamics may affect the outputs for all analyzed controls, especially at the multiples of the rotor frequency (if considered in the controllers)?

**Response**:
The pitch actuator dynamics are not considered in the model. Usually the hydraulic pitch actuator can be modelled very well as a second-order transfer function:

$$\frac{\beta(s)}{\beta_r(s)} = \frac{\omega_n^2}{s^2 + 2 \cdot \zeta \omega_n \cdot s + \omega_n^2} , \tag{8}$$

where $\beta$ and $\beta_r$ denote the pitch angle and the reference pitch angle. $\omega_n$ and $\zeta$ are the natural frequency and the damping factor.

In general, there is no significant influence on the outputs for all analyzed controls in the presence of actuator dynamics under nominal operating conditions. This lies in the fact that in the nominal operating conditions, all the pitch control input and load output signals under multiple rotor frequencies are almost consistent with the original implementation without pitch actuator dynamics. In case a pitch actuator fault occurs, such as changes of the air content or pressure in the hydraulic pitch system, there may exist non-negligible influence of the pitch actuator dynamics on the control algorithm [2,3].
Since this paper mainly focuses on the individual pitch control algorithm under nominal operating conditions and the pitch actuator dynamics will not exert significant influence on the model, the pitch actuator model is hence not included in this study.

[2] Odgaard P., Stoustrup J., Kinnaert M., Fault Tolerant Control of Wind Turbines – a benchmark model, IFAC Proceedings Volumes. 42: 155-160, 2009.

[3] P. F. Odgaard, J. Stoustrup and M. Kinnaert, "Fault-Tolerant Control of Wind Turbines: A Benchmark Model," in IEEE Transactions on Control Systems Technology, vol. 21, no. 4, pp. 1168-1182, July 2013, doi: 10.1109/TCST.2013.2259235.

4. All in all, I may imagine that one may force MBC-IPC (even if integral feedback is present) to have limited ADC, through the imposition of upper boundaries in the cosine and sine control variables (the signals before the anti-Coleman transformation) and through suitable anti-windup filters. If Authors share my opinion, they could note it somewhere in the text (e.g., introduction).

**Response**:
We would like to thank the reviewer for sharing this valuable opinion. We agree that the MPC-IPC approach can be extended to take into account the ADC reduction by manipulating the input signals of the anti-Coleman transformation. In the revised version, we have noted the reviewer's opinion in the introduction section.

**Revised portion**:
(Page: 2-3, Line: 59-61)
Moreover, the Multi-Blade Coordinate (MBC)-based IPC could be also potentially extended to limit ADC by manipulating the input signals of inverse Coleman transformation through suitable wind-up filters. Unfortunately, there are limited numbers of publications investigating such output constrained control strategies.

**Technical corrections:**

1. Abstract. I would smooth the sentence "... the current IPC design not economically viable". It is hard to demonstrate that IPC, considering the entire turbine, is not economically viable.

**Response**:
This sentence has been rephrased in this revised version: ...a decrease of the wind turbine reliability.

**Revised portion**:
(Page: 1, Line: 3-4)
...a decrease of the wind turbine reliability.

2. Line 55: change "to the best of author's knowledge" in "to the best of authors' knowledge".

**Response**:
This sentence has been revised according to the reviewer's comment.

**Revised portion**:
(Page: 2, Line: 55)
..., to the best of authors' knowledge,...

3. Line 62: There is the repetition of the author's name inside and outside the parenthesis. Probably using \citet in latex may give a better output.

**Response**:
In the revised version, all the repetitions of the author's name has been removed according to the reviewer's advice.

**Revised portion**:
(Page: 2)
Line 31: Bossanyi (2003) initially demonstrated...
Line 40: Yang et al. (2011) developed a periodic IPC...
Line 45: ...was proposed by Yang et al. (2015).
Line 53: ...developed by Petrović et al. (2020).

(Page: 3, Line: 66)
...initially proposed by van Wingerden et al. (2011).

(Page: 5, Line: 120)
...the work of Mulders et al. (2019).

4. Figure 2: I believe that the arrow between the "10 MW Wind Turbine (Plant)" and the "Basis functions" is there to emphasize the fact that the sinusoidal are based on the actual azimuth angle coming from the plant. If so, the symbol $\varphi$ can be associated to the arrow itself.

**Response**:
Indeed the sinusoidal are based on the actual azimuth angle measured from the plant, and $\varphi$ should be related to the arrow itself. In the revised version, we have fixed this issue. In addition, the quality of this block diagram has been further improved for demonstrations.

**Revised portion**:
(Page: 6, Figure: 2)

[Figure]

Figure 2. Implementation of cSPRC, which includes online system identification and repetitive control. MPC optimization is used to incorporate the output constraints in repetitive control formulation. $U_f$ represents the basis function, while the symbol $U_f^+$ denotes the Moore-Penrose pseudo-inverse of the basis function. In addition, $u_k$, $y_k$ are the input and output vectors at discrete time index $k$. $\theta_j$, $\tilde{Y}_j$, and $\bar{Y}_j$ denote the transformed control input, transformed control output and the output constraints at rotation $j$. $P$ and $\varphi$ correspond to the period of the disturbance and rotor azimuth.

5. Line 144: change one $\delta u$ in $\delta x$

**Response**:
In the revised version, we have corrected this typo according to the reviewer's comments. Other symbols have been double checked throughout the whole paper.

**Revised portion**:
(Page: 7, Line: 148)
Similarly, $\delta u$, $\delta x$ and $\delta e$ can be defined as well.

6. Line 224: The meaning of $\bar{T}$ is not defined. Could it be $\bar{Y}$? Moreover, in many places it is written that the constraint is defined according to the IEC standards. Those sentences seem a bit vague. How is it practically defined?

**Response**:
$\bar{T}$ is a typo here, and it should be $\bar{Y}$ which represents the output constraint. In the revised version, the error has been fixed. The definition of the output constraint is clarified as follows.

(a) The output constraints can be a user-defined value designed by the wind farm operators, or

(b) In the IEC 61400-1 standard, the safety loads of the wind turbine can be estimated with a safety factor as:
$$\bar{Y} = \xi \cdot Y_c, \tag{9}$$
where $\xi$ is the safety factor and $Y_c$ denotes the characteristic value for the loads, e.g. standard deviation of the loads. For the normal operating condition of the wind turbine, $\xi$ can be selected as 1.35, or

(c) It can be designed according to any other safety limitations and environmental conditions.

In the revised version, the relevant narratives have been reorganized to make it clearer.

**Revised portion**:
(Page: 11-12, Line: 236-244)
The value of the bounds can be determined by the wind farm operator or according to the safety factors of the loads in the design regulation such as IEC 61400-1 (IEC, 2005), or other safety limitations and environmental conditions. For instance, the safety bounds for the blade loads can be define by

$$\bar{Y} = \xi \cdot Y_c, \tag{10}$$

where $\xi$ is the safety factor and $Y_c$ denotes the characteristic value for the loads, e.g. standard deviation of the loads. For the normal operating condition of the wind turbine, $\xi$ can be selected as 1.35 (IEC, 2005).

7. Section 4.2.1: During the simulation, does the wake travel from left to right as it can be inferred by Figure 3? If so, this should be specified in the text to ease the comprehension.

**Response**:
In the simulation, the wake travel from left to right as shown in Figure 3 indeed. Based on the reviewer's suggestion, such a wake propagation has been clearly indicated in the revised paper to ease the comprehension.

**Revised portion**:
(Page: 15, Line: 324-325)
First of all, the wake-rotor interaction is presented in Figure 3. The wind farm wake shed from the upstream turbine propagates from left to right sectors of the rotor, thus leading to the wake-rotor overlap on the downstream turbine.

8. Figure 4: could it be interesting to show also in this case the yawing moment?

**Response**:
The yawing moment of the wind turbine under the same case as in Figure 4 has been presented in the following figure 1 [Response Letter], which indicates similar results as the Out-of-plane bending Moment (MOoP).
In detail, the baseline pitch controller shows maximum yawing moment at around 400s and 750s where the downstream wind turbine is partially overlapped by the wind farm wake. The proposed cSPRC framework is only active in load reduction when the partial wake-rotor overlap cause high aerodynamic loads violating the output constraints during 300s-500s and 700s-800s. On the other hand, the transitional MBC-based IPC aims at a maximum load reduction over the whole period.
As the yawing moment in this figure shows similar patterns as MOoP in Figure 4 of the paper, the yawing moment results are omitted for simplifications.

[Figure]

Figure 1 [Response Letter]. Yawing moment of the wind turbine in LC2 (16m/s wind speed).

9. Figure 4: the caption reads "The steady-state values of MOoP have been removed". This is ok for clarity of the picture, but, at the same time, I may say that in a real environment this cannot be done and, in turn, one has to define variable bounds. In fact, the limitation should be different for different wind speed (or even TI). This should be noted during the mathematical treatment as it could represent an additional complexity in the control scheme.

**Response**:
We agree with the reviewer that in the real implementation this may introduce an additional complexity in the control scheme, and the limitation can be different for different wind states and operating conditions. We would like to clarify that:

(a) In our simulation, a high-pass filter is added to remove the steady-state value of the measured MOoP online, because cSPRC, as an individual pitch control, only aims at the reduction of the load variations. So the steady-state values are not utilized in all the individual pitch control approaches. The high-pass filter is considered as a standard way to remove the steady-state values online.

(b) As the limitation should be designed according to the safety loads of the turbine, the output constraints should be also different for different wind states and operating conditions.

In the revised version, we have noted these points in the mathematical derivations.

**Revised portion**:
(Page: 12, Line: 242-244)
$\bar{Y}$ is usually dependent on the safety loads of the wind turbine, thus such constraints vary for different operating conditions. Furthermore, cSPRC aims at the reduction of MOoP variation, the steady-state values of MOoP are not taken into account in such a IPC approach.

(Page: 15, Line: 330-331)
As IPC strategies only aim at the reduction of MOoP variations, the steady values of MOoP are removed by a high-pass filter.

10. Section 4.2.2: It is not clear whether within the turbulent flow also non-null shear layer is considered.

**Response**:
In the simulations in Section 4.2.2, the turbulent sheared wind flow is generated with the TurbSim model for case study. Therefore the wind shear is also considered for such a turbulent wind flow.
In the revised version, the wind condition has been clearly indicated to make it clearer.

**Revised portion**:
(Page: 14-15, Line: 314-316)
...where the TI is set to be 3.75%. The inflow wind speeds are specified as 12m/s, 16m/s and 20m/s. The wind profile is based on the IEC power-law model (IEC, 2009).

(Page: 18, Section 4.2.2)
**4.2.2 Scenario II**: Turbulent sheared wind condition

(Page: 18, Line: 351)
Another scenario considered in the case study is the turbulent sheared wind condition.

(Page: 1, Line: 9)
wake-rotor overlap and turbulent sheared wind conditions.

(Page: 3, Line: 88-89)
and one where the turbulent sheared wind condition is present, respectively.

(Page: 4, Line: 103)
...turbulent sheared wind scenarios...

(Page: 5, Line: 121-122)
...and turbulent sheared wind flow scenarios.

(Page: 14, Line: 307-308)
(2) Turbulent sheared wind condition: the wind turbine is subjected to turbulent sheared wind flows.

(Page: 14-15, Line: 313-314)
For the turbulent sheared wind condition...

(Page: 15, Table 2)
Turbulent sheared wind case

(Page: 18, Line: 356)
...in the turbulent sheared wind case...

(Page: 18, Line: 360-361)
...both wake overlap and turbulent sheared wind scenarios.

(Page: 20, Table 3)
in turbulent sheared wind conditions (LC4-LC6).

(Page: 21, Line: 395)
where the wake-rotor overlap and turbulent sheared wind conditions...

11. Conclusion: I would expect a comment on multifrequency IPC. Is it possible to easily extend the cSPRC algorithm to multifrequency IPC? Such an extension may be useful for reducing loads in the fixed system, for example the 3p loads at hub, which can be mitigated through 2p and/or 4p IPC.

**Response**:
We agree with the reviewer that the multifrequency IPC will be useful for reducing loads at different frequencies. It is very possible to include multifrequency, *e.g.*, 2P, 3P, etc. into the cSPRC algorithm. In fact, multi-frequencies can be simply considered in this approach by extending the basis function in equation (17), as the basis function is used to project the dynamics onto the frequencies of interests. For instance, both 1P and 2P frequencies can be taken into account if we expand the basis function in equation (17) into:

$$\phi = \underbrace{\begin{bmatrix} \sin(2\pi/P) & \cos(2\pi/P) & \sin(4\pi/P) & \cos(4\pi/P) \\ \sin(4\pi/P) & \cos(4\pi/P) & \sin(8\pi/P) & \cos(8\pi/P) \\ \vdots & \vdots & \vdots & \vdots \\ \sin(2\pi) & \cos(2\pi & \sin(4\pi) & \cos(4\pi) \end{bmatrix}}_{U_f} \otimes I_r \,,$$

where the added third and forth columns will take into account the 2P frequency. Similarly, such an extension can be made for 3P, or even higher frequencies dynamics. In the revised version, a comment on multifrequency IPC has been included.

**Revised portion**:
(Page: 21, Line: 399-400)
Furthermore, cSPRC can be readily extended to a multifrequency IPC by expanding the basis function.